# $\ell_1$ LATENT DISTANCE BASED CONTINUOUS-TIME GRAPH REPRESENTATION

**Zhao-Rong Lai[1], Zheng-Sen Zhou[1*], Liangda Fang[2], Yongsen Zheng[3,4], Ziliang Chen[5]**

[1]Guangdong Key Laboratory of Data Security and Privacy Preserving,
  College of Cyber Security, Jinan University
[2]Department of Computer Science,
  College of Information Science and Technology, Jinan University
[3]College of Computing and Data Science, Nanyang Technological University
[4]Digital Trust Centre, Nanyang Technological University
[5]Research Institute of Multiple Agents and Embodied Intelligence, Peng Cheng Laboratory
`laizhr@jnu.edu.cn, sunchou@stu2022.jnu.edu.cn, fangld@jnu.edu.cn`
`yongsen.zheng@ntu.edu.sg, c.ziliang@yahoo.com`

## ABSTRACT

Continuous-time graph representation (CTGR) is a widely-used methodology in machine learning, physics, bioinformatics, and social networks. The sequential survival process in a latent space with the squared $\ell_2$ distance is an important ultra-low-dimensional embedding for CTGR. However, the squared $\ell_2$ distance violates the triangle inequality, which may cause distortion of the relative node positions in the latent space and thus deteriorates in social, contact, and collaboration networks. Reverting to the $\ell_2$ distance is infeasible because the corresponding integral computation is intractable. To solve these problems, we propose a theoretically-sound $\ell_1$ latent distance based continuous-time graph representation ($\ell_1$LD-CTGR). It facilitates a true latent metric space for the sequential survival process. Moreover, the integral of the hazard function is found to be a closed-form piece-wise exponential integral, which well fits the ultra-low-dimensional embedding. To handle the non-differentiable $\ell_1$ norm, we successfully find a descent direction of the hazard function to replace the gradient, enabling mainstream learning architectures to learn the parameters. Extensive experiments using both synthetic and real-world data show the competitive performance of $\ell_1$LD-CTGR.

## 1 INTRODUCTION

Graph (or network) representations are widely used across various fields, ranging from natural sciences to social sciences, including areas such as particle interactions, biotic activities, and social networking. One of the key tasks in graph representations is to predict the properties of connections and nodes within a graph (Hamilton, 2020; Zhang et al., 2020). Early methods mainly focus on static networks, employing approaches like random walks (Grover & Leskovec, 2016), matrix factorization (Qiu et al., 2019) and Graph Neural Networks (GNNs, (Wu et al., 2021)). More recently, these techniques have been extended to accommodate more complex types of networks, including signed networks (Nakis et al., 2023) and hierarchical structures (Nakis et al., 2024a).

Compared with static networks, dynamic networks can better reveal intrinsic patterns and interactions between nodes over time (Çelikkanat et al., 2024). At first, researchers focuses on discrete-time networks, using probabilistic models (Heaukulani & Ghahramani, 2013), multiscale structures (Herlau et al., 2013), and hyperbolic space (Yang et al., 2021). More recently, continuous-time networks have emerged, modeled by methods like random walks (Zuo et al., 2018; Nguyen et al., 2018), Poisson processes (Fan et al., 2021; Çelikkanat et al., 2022), Hawkes processes (Hawkes, 1971; Arastuie et al., 2020; Huang et al., 2022), and sequential survival processes (Çelikkanat et al., 2024). However, Poisson and Hawkes processes treat network connections as instantaneous events, largely ignoring the persistence of edges in the likelihood function. To address this limitation, the

---

*Corresponding author.

Graph Sequential Survival Process (GRASSP (Çelikkanat et al., 2024)) is proposed, which dynamically captures both the existence and absence of connections. Since survival analysis is effective at modeling the duration leading up to the failure or termination of an entity, it is also suited for modeling the lifespan of connections between nodes, which is particularly effective in social, contact, and collaboration networks.

The distance between two nodes in the latent space plays a crucial role in determining their likelihood of being connected. Nodes are more likely to connect when they are closer in the latent space, while they are more likely to disconnect when farther apart. This property can be captured using a latent-distance-based hazard function within the sequential survival process (Çelikkanat et al., 2024). A common choice for latent distance is the **squared $\ell_2$ distance**, which is rooted in the general real-world Euclidean geometry. It well fits the ultra-low-dimensional embedding (Çelikkanat et al., 2024; Nakis et al., 2024b; Huang et al., 2024a), which serves as a dimensionality reduction scheme, improves the conciseness of graph representation learning, and saves computational cost (Hamilton, 2020; Xu, 2021; Barros et al., 2021). Moreover, the integral of the corresponding hazard function is the well-studied Gaussian integral, commonly used in probability and statistics.

However, **the squared $\ell_2$ distance violates the triangle inequality (see Eq. 8), thus it is not a valid distance for the latent metric space.** The corresponding hazard function may distort the relative node positions in the latent space, and thus deteriorates in social, contact, and collaboration networks. For example, to determine whether A has a closer relationship with B than C in this latent space, the triangle inequality should not be violated. This problem cannot be fixed by reverting to the $\ell_2$ distance (without a square), or using other $\ell_p$ distance with $p > 1$ and $p \neq 2$, because the integral computation of the corresponding hazard function is intractable. Some recent works indicate that the $\ell_1$ distance can serve as an alternative of the $\ell_2$ distance when the latter becomes ineffective in certain scenarios (Lai & Wang, 2024; Çelikkanat et al., 2024; Wang et al., 2025). It motivates us to use the $\ell_1$ distance to facilitate a true latent metric space for the sequential survival process. We find that the integral of the hazard function with $\ell_1$ distance is a closed-form piece-wise exponential integral, which well fits the ultra-low-dimensional embedding.

In this paper, we propose a theoretically-sound $\ell_1$ latent distance based continuous-time graph representation ($\ell_1$**LD-CTGR**). Our **main contributions** can be summarized as follows:

**1.** We verify that the average $\ell_1$ distance during an event time can be bounded by two values that are determined by the connected and disconnected states and the survival function, which is similar to the bounded property of the squared $\ell_2$ distance case.
**2.** We derive the integral of the hazard function for $\ell_1$LD-CTGR, which is a closed-form piece-wise exponential integral, markedly different from the Gaussian integral of the squared $\ell_2$ distance case.
**3.** We propose an efficient integral computing algorithm for the ultra-low-dimensional embedding of $\ell_1$LD-CTGR.
**4.** The $\ell_1$ distance, the corresponding hazard function, and the survival function are all non-differentiable. Hence we find a descent direction to replace the gradient, allowing mainstream learning architectures to learn the parameters of the graph.

## 2 RELATED WORKS AND UNSOLVED PROBLEMS

We introduce the preliminary framework based on (Çelikkanat et al., 2024), then raise some problems of the currently-used squared $\ell_2$ distance.

### 2.1 CONTINUOUS-TIME GRAPH REPRESENTATION

Denote $\mathcal{G} := (\mathcal{V}, \mathcal{E})$ as a graph, where $\mathcal{V} := \{1, 2, \cdots, N\}$ represents the vertex set, and $\mathcal{E} := \cup_{i,j \in \mathcal{V}} \mathcal{E}_{ij}$ represents the edge set. Here, $\mathcal{E}_{ij} := (i, j) \in \mathcal{V}^2$ denotes that nodes $i$ and $j$ are connected by an edge. In the continuous-time interval scenario, the time interval $[T] := [0, T)$ must also be considered. Correspondingly, a connection between nodes $i$ and $j$ may last from $t_k \in [T]$ to $t_{k+1} \in [T]$. Therefore, the edge set should be extended to the following set of tuples $\mathcal{E} := \{(i, j, t_k, t_{k+1}) \in \mathcal{V}^2 \times [T]^2 : t_k < t_{k+1}\}$, where $(i, j, t_k, t_{k+1})$ and $(i, j, t_l, t_{l+1})$ represent two distinct connections if $[t_k, t_{k+1}) \cap [t_l, t_{l+1}) = \emptyset$.

An event time $e_m$ is defined as the time when an edge gets connected or disconnected, with $e_0 = 0$ always being an event time. If a node pair undergoes $M$ events $e_0 = 0 < e_1 < e_2 < \cdots < e_{M-1} < T$, then there are $M$ consecutive intervals $\{[e_m, e_{m+1}]\}_{m=0}^{M-1}$, each with different states. The state

function $s : \mathcal{V}^2 \times [T] \to \mathcal{S}$ with $\mathcal{S} := \{1, -1\}$ indicates whether an edge $(i, j)$ is connected $(s = 1)$ or disconnected $(s = -1)$ at time $t$.

A survival function describes how long an object can exist or function until it fails or dies:

$$S(t) := \mathbb{P}\{\mathcal{T} > t\} = \exp\left(-\int_0^t \lambda(\tau)\,\mathrm{d}\tau\right), \tag{1}$$

where $\lambda : \mathbb{R}^+ := [0, +\infty) \to \mathbb{R}^+$ denotes the hazard function for this survival process, and $\mathcal{T}$ represents the lifetime random variable. Thus, the "surviving" and "dying" states of a connection can be characterized as part of a sequential survival process. Let $s_0 \in \mathcal{S}$ be the initial state and $\{M(t)\}_{t \geqslant 0}$ be a counting process that records the number of events up to time $t$. The probability function of $M(t)$ equaling $m$ is given by

$$\mathbb{P}\{M(t) = m\} = \int_{\xi \in \mathscr{D}} \prod_{k=1}^m \frac{\int_{e_{k-1}}^{e_k} \lambda(s_k, \tau)\,\mathrm{d}\tau}{\exp\left(\int_{e_{k-1}}^{e_k} \lambda(s_k, \tau)\,\mathrm{d}\tau\right)}\,\mathrm{d}\xi, \tag{2}$$

where $\mathscr{D} := \{(t_1, t_2, \cdots, t_m) \in [T]^m : 0 \leqslant t_1 < t_2 < \cdots < t_m < T\}$ denotes the integration domain, and $\lambda(s_k, \tau)$ represents the hazard rate at state $s_k$ and time $\tau$. The integration variables $(t_1, t_2, \cdots, t_m)$ depend on the event times $(e_1, e_2, \cdots, e_m)$. According to the fundamental theorem of calculus, the density function for $\mathbb{P}\{M(t) = m\}$ at $(e_1, e_2, \cdots, e_m)$ is:

$$p_{\{M(t)=m\}}(e_1, e_2, \cdots, e_m) = \prod_{k=1}^m \frac{\lambda(s_k, e_k)}{\exp\left(\int_{e_{k-1}}^{e_k} \lambda(s_k, \tau)\,\mathrm{d}\tau\right)}. \tag{3}$$

Next, we develop a suitable hazard function $\lambda(s, t)$ to control the connections between nodes. This function is embedded in a latent metric space $(\mathrm{X}, \mathfrak{d}_\mathrm{X})$, where $\mathfrak{d}_\mathrm{X}$ denotes the distance function. The position of node $i$ at time $t$ is represented by a point $\boldsymbol{r}_i(t)$ in X. To model the connection based on the distance between nodes, $\lambda_{ij}(s, t)$ can be defined as

$$\lambda_{ij}(s, t) := \psi^s(\mathfrak{d}_\mathrm{X}(\boldsymbol{r}_i(t), \boldsymbol{r}_j(t))), \tag{4}$$

where $\psi^s : \mathbb{R} \to \mathbb{R}^+$ is a continuous function. Then the log-likelihood function of the entire graph is given by:

$$\log p(\mathcal{G}|\Omega) = \sum_{(i,j) \in \mathcal{V}^2} \sum_{k=1}^{|\mathcal{E}_{ij}|} \left(\log \lambda_{ij}(s_k, e_k) - \int_{e_{k-1}}^{e_k} \lambda_{ij}(s_k, \tau)\,\mathrm{d}\tau\right), \tag{5}$$

where $\Omega$ represents the set of hyperparameters. Recent attention focuses on the ultra-low-dimensional embedding where the latent space $\mathrm{X} := \mathbb{R}^{\mathcal{D}}$ has a very low dimensionality $\mathcal{D}$, ideally $\mathcal{D} = 2$ (Çelikkanat et al., 2024; Nakis et al., 2024b; Huang et al., 2024a). It reduces dimensionality directly, improves the conciseness of graph representation learning, and saves computational cost (Hamilton, 2020; Xu, 2021; Barros et al., 2021).

## 2.2 PROBLEMS OF SQUARED $\ell_2$ DISTANCE

In GRASSP (Çelikkanat et al., 2024), the following specific form for $\lambda_{ij}(s, t)$ is proposed:

$$\lambda_{ij}(s, t) := \exp\left(\beta(s) + s\|\boldsymbol{r}_i(t) - \boldsymbol{r}_j(t)\|_2^2\right), \tag{6}$$

where the parameter $\beta(s)$ captures global connectedness (Krivitsky et al., 2009; Çelikkanat et al., 2022), and the squared $\ell_2$ distance $\|\cdot\|_2^2$ has a simple computational form:

$$\|\boldsymbol{r}\|_2^2 = \boldsymbol{r}^\top \boldsymbol{r}, \quad \nabla(\|\boldsymbol{r}\|_2^2) = 2\boldsymbol{r}. \tag{7}$$

This advantage makes GRASSP efficient and effective in the ultra-low-dimensional embedding $\mathcal{D} = 2$. **However, $\|\cdot\|_2^2$ violates the triangle inequality, thus it is not a valid distance for the latent space. Specifically,**

$$\|\boldsymbol{r}_i - \boldsymbol{r}_j\|_2^2 \nleqslant \|\boldsymbol{r}_i - \boldsymbol{r}_\mathfrak{k}\|_2^2 + \|\boldsymbol{r}_\mathfrak{k} - \boldsymbol{r}_j\|_2^2, \ \exists\, \boldsymbol{r}_i, \boldsymbol{r}_j, \boldsymbol{r}_\mathfrak{k}. \tag{8}$$

**A simple counterexample is that $(\boldsymbol{r}_i - \boldsymbol{r}_\mathfrak{k})^\top (\boldsymbol{r}_\mathfrak{k} - \boldsymbol{r}_j) > 0$. Due to this problem, $\lambda_{ij}(s, t)$ in (6) cannot determine whether node $\mathfrak{k}$ is closer to node $i$ than node $j$, which may cause distortion in the latent metric space $(\mathrm{X}, \mathfrak{d}_\mathrm{X})$ and affect the performance in social, contact, and collaboration networks.** To better see this distortion, we can divide the interval $[-1, 1]$ into $Q$ segments and consider a set of nodes $\boldsymbol{r}_q = (-1 + \frac{2q}{Q}, 0) \in \mathbb{R}^2$, $q = 0, 1, ..., Q$. Then the indirect distance between

$r_0$ and $r_Q$ is

$$\sum_{q=1}^{Q} \|r_q - r_{q-1}\|_2^2 = Q \cdot (\frac{2}{Q})^2 = \frac{4}{Q} \to 0 \quad \text{as} \quad Q \to \infty. \tag{9}$$

It indicates that adding infinite nodes between $r_0$ and $r_Q$ can shrink their indirect distance to nearly zero, which is counter-intuitive and unconventional. This may affect the local geometrical structure and relative node positions in the latent space.

One may argue that this problem can be fixed by reverting to the $\ell_2$ distance without a square. However, **using the $\ell_2$ distance makes it intractable and inaccurate to compute the integral** $\int_{e_{k-1}}^{e_k} \lambda_{ij}(s_k, \tau) \, d\tau$ **in the log-likelihood** (5). Following the same variable change technique as (16), the core computation for the $\ell_2$ distance case is:

$$\int_0^{e_u - e_l} \exp\left(s\|\Delta x_{ij} + \Delta v_{ij}t\|_2\right) \, dt. \tag{10}$$

There is no closed-form solution to this integral, thus it should be computed via numerical methods. This causes two main drawbacks: **1)** The interval $[0, e_u - e_l]$ should be divided into a sufficiently large number $H$ of subintervals with length $\Delta t$, then $\mathcal{O}(\mathcal{D})$ is required to conduct the summation and $\ell_2$ norm in **each** of the $H$ subintervals. Last, $(H + 1)$ exponents are evaluated to compute the entire integral (the complexity of an exponential operation can be tens of that of a multiplication). Thus the total computational complexity for the $\ell_2$ distance integral is $\mathcal{O}(H\mathcal{D})$, which can be hundreds or even thousands of $\tilde{\mathcal{O}}(\mathcal{D})$ for both GRASSP and $\ell_1$LD-CTGR (see the last paragraph of Section 3.4 and Table A1). **2)** The numerical integral involves an approximation error compared to the exact integral, which can be amplified by composite function relationships. According to Simpson's Rule with an even $H$, the numerical integral for (10) is

$$\mathcal{I}_{\ell_2} := \frac{\Delta t}{3}\left[\exp\left(s\|\Delta x_{ij}\|_2\right) + 4 \cdot \sum_{\substack{h=1, \\ h \text{ is odd}}}^{H-1} \exp\left(s\|\Delta x_{ij} + \Delta v_{ij}(h\Delta t)\|_2\right)\right.$$

$$\left. + 2 \cdot \sum_{\substack{h=2, \\ h \text{ is even}}}^{H-2} \exp\left(s\|\Delta x_{ij} + \Delta v_{ij}(h\Delta t)\|_2\right) + \exp\left(s\|\Delta x_{ij} + \Delta v_{ij}(H\Delta t)\|_2\right)\right]. \tag{11}$$

It yields an approximation error of $O((\Delta t)^4)$. For an integration interval $[0, 1]$, $H = 1000$ leads to $\Delta t = 10^{-3}$ and an approximation error of $O(10^{-12})$. However, even such an approximating accuracy cannot bring about good performance in our task (see Table A4). In contrast, $\int_{e_{k-1}}^{e_k} \lambda_{ij}(s_k, \tau) \, d\tau$ using the squared $\ell_2$ distance is a well-studied exact Gaussian integral, as explained in the following subsection.

## 2.3 GAUSSIAN INTEGRAL INDUCED BY SQUARED $\ell_2$ DISTANCE

To model the continuous-time movement of a node, a piece-wise linear approximation with $B$ bins can be used:

$$r_i(t) = x_i^{(0)} + \Delta_B \sum_{b=1}^{\lfloor t/\Delta_B \rfloor} v_i^{(b)} + \text{mod}(t, \Delta_B)v_i^{(\lfloor t/\Delta_B \rfloor + 1)}, \tag{12}$$

where $x_i^{(0)}$ represents the initial position, $\Delta_B := T/B$ denotes the bin width, and $v_i^{(b)}$ is the velocity for the $b$-th bin. The integral of $\lambda_{ij}(s, t)$ in (6) over an interval $(e_l, e_u)$ can be computed as:

$$\int_{e_l}^{e_u} \exp\left(\beta(s) + s\|r_i(t) - r_j(t)\|_2^2\right) \, dt$$

$$= \frac{\sqrt{\pi}}{2\|\Delta v_{ij}\|_2} \exp\left(\beta(s) + s\|\Delta x_{ij}\|_2^2 - s\rho_{ij}^2\right) \cdot E_{ij}(s, \tau(e_l), \tau(e_u)),$$

$$E_{ij}(s, \tau(e_l), \tau(e_u)) := \begin{cases} \text{erf}(\tau(e_u)) - \text{erf}(\tau(e_l)) & \text{if } s = 1 \\ \text{erfi}(\tau(e_u)) - \text{erfi}(\tau(e_l)) & \text{if } s = -1 \end{cases},$$

$$\tau(t) := \|\Delta v_{ij}\|_2 t + \rho_{ij}, \quad \Delta x_{ij} := r_i(e_l) - r_j(e_l),$$

$$\Delta \boldsymbol{v}_{ij} := \boldsymbol{v}_i(e_l) - \boldsymbol{v}_j(e_l), \quad \rho_{ij} := \Delta \boldsymbol{x}_{ij}^\top \Delta \boldsymbol{v}_{ij} / \|\Delta \boldsymbol{v}_{ij}\|_2, \tag{13}$$

where $\mathrm{erf}$ and $\mathrm{erfi}$ denote the error function and the imaginary error function, respectively, for the Gaussian distribution. Both functions are well-studied Gaussian integrals that can be conveniently computed by sophisticated numerical methods. This is a main reason why the squared $\ell_2$ distance is preferred to the $\ell_2$ distance. As for other $\ell_p$ distance with $p > 1$ and $p \neq 2$, the corresponding integral is still intractable.

## 3 METHODOLOGY

**We propose the following $\ell_1$ distance based hazard function**:
$$\lambda_{ij}(s,t) := \exp\left(\beta(s) + s\|\boldsymbol{r}_i(t) - \boldsymbol{r}_j(t)\|_1\right), \tag{14}$$
where $\|\cdot\|_1$ represents the $\ell_1$ norm. **It drops the square of the distance and facilitates a true latent metric space $(\mathrm{X}, \mathfrak{d}_\mathrm{X})$ with $\mathfrak{d}_\mathrm{X} := \|\cdot\|_1$, whose integral is tractable and found to be a closed-form piece-wise exponential integral.** By this means, we can solve the problems of latent space distortion and integral computation in Section 2.2. **To handle (14), we need to reform the entire sequential survival process and overcome several challenges**: **1.** The signs of different dimensions for $(\boldsymbol{r}_i(t) - \boldsymbol{r}_j(t))$ may change asynchronously within the interval $[e_l, e_u)$, which complicates the integral computation $\int_{e_l}^{e_u} \lambda_{ij}(s,t)\,\mathrm{d}t$. **2.** An efficient integral computing scheme should be developed for the ultra-low-dimensional embedding to save the computational cost. **3.** $\lambda_{ij}(s,t)$ is non-differentiable with respect to (w.r.t.) $\boldsymbol{r}_i$ or $\boldsymbol{r}_j$, which could hinder the learning of model parameters.

### 3.1 BOUNDEDNESS OF AVERAGE $\ell_1$ DISTANCE

An important property in the sequential survival process is that the connected and disconnected states of a node pair, together with the survival function determine the lower and upper bounds of the corresponding distance (Çelikkanat et al., 2024). We verify that this property also holds in the $\ell_1$ distance case, presented by the following proposition.

**Proposition 1.** *Let $e_k$ and $e_{k+1}$ be two consecutive event times for a node pair $(i, j)$. Then the average $\ell_1$ distance of this node pair in the interval $[e_k, e_{k+1})$ with survival function $S_{ij}(\cdot)$ and state $s \in \{1, -1\}$ can be bounded by:*

$$\mathcal{B}(-1) \leqslant \frac{1}{e_{k+1} - e_k} \int_{e_k}^{e_{k+1}} \|\boldsymbol{r}_i(t) - \boldsymbol{r}_j(t)\|_1 \,\mathrm{d}t \leqslant \mathcal{B}(1),$$
$$\mathcal{B}(s) := s(\log(\log(S_{ij}(e_k)) - \log(S_{ij}(e_{k+1}))) - \beta(s) - \log(e_{k+1} - e_k)). \tag{15}$$

The proof is given in Appendix A.1. This proposition indicates that the average $\ell_1$ distance of a node pair $(i, j)$ during time span $[e_k, e_{k+1})$ is controllable by the state $s$ in a practical sense. The left inequality indicates that the average $\ell_1$ distance should be no less than $\mathcal{B}(-1)$ in the disconnected state, while the right inequality indicates that the average $\ell_1$ distance should be no larger than $\mathcal{B}(1)$ in the connected state. In other words, nodes $i$ and $j$ cannot get too close to (or too far from) each other to maintain a disconnected (or connected) state. This aligns with the general understanding of nature and society.

### 3.2 INTEGRAL COMPUTATION

The integral of the hazard function $\int_{e_l}^{e_u} \lambda_{ij}(s,t)\,\mathrm{d}t$ is a key component in the maximum log-likelihood estimation (MLE) as given in (5). To compute this, we employ a piece-wise linear approximation (12) similar to the $\ell_2$ distance version in (13).

$$\int_{e_l}^{e_u} \lambda_{ij}(s,t)\,\mathrm{d}t = \int_{e_l}^{e_u} \exp\left(\beta(s) + s\|\boldsymbol{r}_i(t) - \boldsymbol{r}_j(t)\|_1\right)\,\mathrm{d}t$$
$$= \int_{e_l}^{e_u} \exp\left(\beta(s) + s\|\Delta\boldsymbol{x}_{ij} + \Delta\boldsymbol{v}_{ij}(t - e_l)\|_1\right)\,\mathrm{d}t$$
$$= \exp\left(\beta(s)\right) \int_0^{e_u - e_l} \exp\left(s\|\Delta\boldsymbol{x}_{ij} + \Delta\boldsymbol{v}_{ij}t\|_1\right)\,\mathrm{d}t, \tag{16}$$

where $\Delta \boldsymbol{x}_{ij}(e_l) := \boldsymbol{r}_i(e_l) - \boldsymbol{r}_j(e_l)$ and $\Delta \boldsymbol{v}_{ij}(e_l) := \boldsymbol{v}_i(e_l) - \boldsymbol{v}_j(e_l)$ represent the instantaneous relative position and velocity of the node pair $(i, j)$. These quantities are recorded at each key time $e_l$ and used to compute the integral up to $e_u$. Since $\exp(\beta(s))$ is a constant w.r.t. the integral over $t$, it can be put outside this integral. Then the key work lies in computing the latter part integral of (16). The main result is summarized as the following theorem, while the detailed computation and proof are provided in Appendix A.2.

**Theorem 1.**

$$\int_{e_l}^{e_u} \lambda_{ij}(s,t)\,\mathrm{d}t = \begin{cases} \exp(\beta(s))\,\mathcal{I}_{ij,0,e_u-e_l} & \text{if } z_{ij,(\underline{c})} > e_u - e_l \text{ or } z_{ij,(\overline{c})} < 0 \\ \exp(\beta(s))\left(\mathcal{I}_{ij,0,(\underline{c})} + \sum_{c=\underline{c}}^{\overline{c}-1}\mathcal{I}_{ij,(c),(c+1)} + \mathcal{I}_{ij,(\overline{c}),e_u-e_l}\right) & \text{else} \end{cases}, \quad (17)$$

where $z_{ij,(\underline{c})}$ and $z_{ij,(\overline{c})}$ denote the zero points defined in (34) and (35). $\mathcal{I}_{ij,(c),(c+1)}$, $\mathcal{I}_{ij,0,(\underline{c})}$, and $\mathcal{I}_{ij,(\overline{c}),e_u-e_l}$ are integral pieces defined in (37), (38), (40), and (41), respectively.

Theorem 1 indicates that $\int_{e_l}^{e_u}\lambda_{ij}(s,t)\,\mathrm{d}t$ is a closed-form piece-wise exponential integral in the $\ell_1$ distance case, which is very different from the Gaussian integral (13) in the $\ell_2$ distance case. Moreover, $\mathcal{I}_{ij,(c),(c+1)}$ can be computed successively and conveniently from $c = \underline{c}$ to $c = \overline{c} - 1$ with slight updates of some intermediate variables in (44). We demonstrate the application of Theorem 1 with a simple example in Appendix A.2.

## 3.3 Ultra-low-dimensional Embedding

The ultra-low-dimensional embedding has been catching attention recently (Çelikkanat et al., 2024; Nakis et al., 2024b; Huang et al., 2024a), since it reduces dimensionality by its nature, improves the conciseness of graph representation learning, and saves computational cost (Hamilton, 2020; Xu, 2021; Barros et al., 2021). As for $\ell_1$LD-CTGR, we find that there are at most 2 zero points in the ultra-low-dimensional embedding with $\mathcal{D} = 2$, which are compatible with the two limits $e_k$ and $e_{k+1}$ of integration. By this means, we can develop an efficient integral computing algorithm in a tensor-parallelized form.

**Theorem 2.** *In the ultra-low-dimensional embedding with $\mathcal{D} = 2$, the integrals $\left\{\int_{e_k}^{e_{k+1}}\lambda_{ij}(s,t)\,\mathrm{d}t\right\}_{i,j,k}$ can be computed in a tensor-parallelized way:*

$$\left[\int_{e_k}^{e_{k+1}}\lambda_{ij}(s,t)\,\mathrm{d}t\right]_{i,j,k} = \exp(\boldsymbol{\beta}(\boldsymbol{s})) \odot (\boldsymbol{\mathcal{I}}_1 + \boldsymbol{\mathcal{I}}_2 + \boldsymbol{\mathcal{I}}_3), \quad (18)$$

*where $\boldsymbol{\mathcal{I}}_1, \boldsymbol{\mathcal{I}}_2, \boldsymbol{\mathcal{I}}_3 \in \mathbb{R}^{L \times 1}$ are tensor-parallelized integral pieces defined in (58), (59), and (60), respectively. $\odot$ denotes the element-wise multiplication. $\boldsymbol{\beta}(\boldsymbol{s}) \in \mathbb{R}^{L \times 1}$ and $\boldsymbol{s} \in \{1, -1\}^{L \times 1}$. The dimensionality of the above tensors is the same as the total number of observations $L := |\{(i,j,k)\}|$.*

The proof is given in Appendix A.3, which also demonstrates the entire computing process for $\boldsymbol{\mathcal{I}}_1$, $\boldsymbol{\mathcal{I}}_2$, and $\boldsymbol{\mathcal{I}}_3$. By this means, $\ell_1$LD-CTGR not only well fits the ultra-low-dimensional embedding like GRASSP, but also facilitates a true latent metric space.

## 3.4 Descent Direction and Subgradient Computation

There are 3 primary sets of learnable parameters in the proposed $\ell_1$LD-CTGR model: the initial positions $\boldsymbol{x}^{(0)} \in \mathbb{R}^{N \cdot \mathcal{D}}$, the velocities for the piece-wise linear approximation bins $\boldsymbol{v} \in \mathbb{R}^{N \cdot B \cdot \mathcal{D}}$, and the global connectedness parameters $[\beta(s=1), \beta(s=-1)]^\top \in \mathbb{R}^2$. Each of the $N$ nodes is represented as a point in a $\mathcal{D}$-dimensional latent space, resulting in a total of $N \cdot \mathcal{D}$ parameters for $\boldsymbol{x}^{(0)}$. Additionally, with $B$ bins per node and different velocities for each bin, there are $N \cdot B \cdot \mathcal{D}$ parameters for $\boldsymbol{v}$ in total (see Eq. 12).

The learning process involves minimizing the negative log-likelihood $-\log p(\mathcal{G}|\Omega)$ defined in (5) w.r.t. $\boldsymbol{x}^{(0)}$, $\boldsymbol{v}$, and $\beta(s)$. This is typically achieved through gradient descent. Specifically, we need to compute the following partial derivatives:

$$\frac{\partial \log \lambda_{ij}(s_k, e_k)}{\partial \phi} = \frac{1}{\lambda_{ij}(s_k, e_k)} \cdot \frac{\partial \lambda_{ij}(s_k, e_k)}{\partial \phi}, \frac{\partial\left[\int_{e_k}^{e_{k+1}}\lambda_{ij}(s,t)\,\mathrm{d}t\right]}{\partial \phi} = \int_{e_k}^{e_{k+1}}\left[\frac{\partial \lambda_{ij}(s,t)}{\partial \phi}\right]\mathrm{d}t,$$

(19)

where $\phi = \boldsymbol{x}^{(0)}$, $\boldsymbol{v}$, or $\beta(s)$, respectively. In both terms, $\frac{\partial \lambda_{ij}(s,t)}{\partial \phi}$ must exist. While $\frac{\partial \lambda_{ij}(s,t)}{\partial \boldsymbol{\beta}}$ does exist, $\frac{\partial \lambda_{ij}(s,t)}{\partial \boldsymbol{x}^{(0)}}$ and $\frac{\partial \lambda_{ij}(s,t)}{\partial \boldsymbol{v}}$ may not. Since this subsection focuses on the primary parameters, we temporarily omit the variables $(s,t)$ of $\lambda_{ij}$ for clarity.

Mainstream learning architectures, such as Pytorch[1], rely on backpropagation, which essentially applies the chain rule for gradient computation. Combining (12) and (14), we have

$$\frac{\partial \lambda_{ij}}{\partial \boldsymbol{x}_i^{(0)}} = \frac{\partial \lambda_{ij}}{\partial \boldsymbol{r}_i} \cdot \frac{\partial \boldsymbol{r}_i}{\partial \boldsymbol{x}_i^{(0)}} = \frac{\partial \lambda_{ij}}{\partial \boldsymbol{r}_i} \cdot \boldsymbol{I} = \frac{\partial \lambda_{ij}}{\partial \boldsymbol{r}_i}, \frac{\partial \lambda_{ij}}{\partial \boldsymbol{v}_i^{(b)}} = \frac{\partial \lambda_{ij}}{\partial \boldsymbol{r}_i} \cdot \frac{\partial \boldsymbol{r}_i}{\partial \boldsymbol{v}_i^{(b)}} = \frac{\partial \lambda_{ij}}{\partial \boldsymbol{r}_i} \cdot \Delta_B \boldsymbol{I} = \Delta_B \cdot \frac{\partial \lambda_{ij}}{\partial \boldsymbol{r}_i}, \quad (20)$$

where $\boldsymbol{I}$ denotes the identity matrix with its dimensionality inferred from the context. In the squared $\ell_2$ distance form (6), the gradient $\frac{\partial \lambda_{ij}}{\partial \boldsymbol{r}_i}$ exists, making backpropagation valid. However, in the $\ell_1$ distance form (14), $\frac{\partial \lambda_{ij}}{\partial \boldsymbol{r}_i}$ does not exist due to the non-differentiable nature of the $\ell_1$ norm, making backpropagation invalid.

To address this issue, we need to find a descent direction of $\lambda_{ij}$ w.r.t. $\boldsymbol{r}_i$ to replace the negative gradient $-\frac{\partial \lambda_{ij}}{\partial \boldsymbol{r}_i}$, which leads to the concept of subgradient.

**Definition 1** (The Fréchet Subdifferential). For a proper lower semicontinuous function $h : \mathbb{R}^{\mathcal{D}} \to \mathbb{R} \cup \{+\infty\}$, the subgradient set at a point $\boldsymbol{r} \in \mathrm{dom}\, h$ is defined as follows:

$$\partial h(\boldsymbol{r}) := \left\{ \boldsymbol{w} \in \mathbb{R}^{\mathcal{D}} : \liminf_{\substack{\boldsymbol{y} \to \boldsymbol{r} \\ \boldsymbol{y} \neq \boldsymbol{r}}} \frac{h(\boldsymbol{y}) - h(\boldsymbol{r}) - \boldsymbol{w}^{\top}(\boldsymbol{y} - \boldsymbol{r})}{\|\boldsymbol{y} - \boldsymbol{r}\|_2} \geqslant 0 \right\}. \quad (21)$$

In brief, a subgradient $\boldsymbol{w}$ (if it exists) represents an ascent direction of $h$, possessing similar properties to a gradient (Lin et al., 2024). In our task, it is well-known that $\mathrm{sign}(\boldsymbol{r}) \in \partial \|\boldsymbol{r}\|_1$. Therefore, we can directly set $\partial \|\boldsymbol{r}\|_1 := \mathrm{sign}(\boldsymbol{r})$. By slightly abusing the use of chain rule, we further obtain the following ascent direction:

$$\partial \lambda_{ij}(\boldsymbol{r}_i) := \exp\left(\beta(s) + s\|\boldsymbol{r}_i(t) - \boldsymbol{r}_j(t)\|_1\right) \cdot \partial(s\|\boldsymbol{r}_i(t) - \boldsymbol{r}_j(t)\|_1)$$
$$= \exp\left(\beta(s) + s\|\boldsymbol{r}_i(t) - \boldsymbol{r}_j(t)\|_1\right) \cdot s \cdot \mathrm{sign}(\boldsymbol{r}_i(t) - \boldsymbol{r}_j(t)). \quad (22)$$

**Theorem 3.** *Let $\partial \lambda_{ij}(\boldsymbol{r}_i)$ be defined in (22), and let $\mathcal{C} := \{d \in \mathcal{D} : r_{i,d} = r_{j,d}\}$ be the index set of dimensions where nodes $i$ and $j$ have the same coordinate values.*

1. *If $\mathcal{C} = \mathcal{D}$, then $\lambda_{ij}$ is differentiable at $\boldsymbol{r}_i$ and $\partial \lambda_{ij}(\boldsymbol{r}_i) = \nabla \lambda_{ij}(\boldsymbol{r}_i) = \boldsymbol{0}$.*

2. *If $\mathcal{C} \subsetneq \mathcal{D}$, then $-\partial \lambda_{ij}(\boldsymbol{r}_i)$ is a descent direction of $\lambda_{ij}$ at $\boldsymbol{r}_i$.*

3. *If $\mathcal{C} = \emptyset$, or $\mathcal{C} \neq \emptyset$ and $s = 1$, then $\partial \lambda_{ij}(\boldsymbol{r}_i)$ is a subgradient of $\lambda_{ij}$ at $\boldsymbol{r}_i$.*

The proof is given in Appendix A.4. This theorem indicates that in all cases, $-\partial \lambda_{ij}(\boldsymbol{r}_i)$ in (22) characterizes either a stationary point or a descent direction. Particularly in Item 3, $\partial \lambda_{ij}(\boldsymbol{r}_i)$ becomes a subgradient. However, when $s = -1$, $\partial \lambda_{ij}(\boldsymbol{r}_i)$ is not a subgradient as demonstrated in Case 4 of the proof in Appendix A.4, because

$$\frac{\exp(-\sum_{d=1}^{\mathcal{D}} |y_d - r_{i,d}|) - 1 + \sum_{d \notin \mathcal{C}} |y_d - r_{i,d}|}{\sqrt{\sum_{d=1}^{\mathcal{D}} (y_d - r_{i,d})^2}} \not\geqslant 0 \quad (23)$$

in (72). To see this, we consider a counterexample with $\mathcal{D} = \{1, 2\}$, $\mathcal{C} = \{1\}$ and $y_d - r_{i,d} = \epsilon > 0$ for $d = 1, 2$. Then (23) becomes

$$\frac{\exp(-2\epsilon) - 1 + \epsilon}{\sqrt{2}\epsilon} = \frac{-2\epsilon + o(-2\epsilon) + \epsilon}{\sqrt{2}\epsilon} = -\frac{1}{\sqrt{2}} - o(\sqrt{2}) < 0 \quad (24)$$

as $\epsilon \to 0$. This violates the inequality in (21), which means that $\partial \lambda_{ij}(\boldsymbol{r}_i)$ is not a subgradient. Nevertheless, $-\partial \lambda_{ij}(\boldsymbol{r}_i)$ is still a descent direction of $\lambda_{ij}$ at $\boldsymbol{r}_i$ in this case.

To summarize this subsection, we can directly compute $\partial \lambda_{ij}(\boldsymbol{r}_i)$ by (22) and check whether $\partial \lambda_{ij}(\boldsymbol{r}_i) = \boldsymbol{0}$ (which is equivalent to $\mathcal{C} = \mathcal{D}$). If so, then $\partial \lambda_{ij}(\boldsymbol{r}_i) = \nabla \lambda_{ij}(\boldsymbol{r}_i) = \boldsymbol{0}$. If not,

---

[1] https://pytorch.org/

then $-\partial\lambda_{ij}(\boldsymbol{r}_i)$ is a descent direction of $\lambda_{ij}$ at $\boldsymbol{r}_i$, and a line search method can be applied to reduce $\lambda_{ij}$. In this way, $\partial\lambda_{ij}(\boldsymbol{r}_i)$ is compatible with mainstream learning architectures like Pytorch for the proposed $\ell_1$LD-CTGR.

As for computational complexity, the negative log-likelihood of $\ell_1$LD-CTGR is nonconvex, which requires $\mathcal{O}(1/\epsilon^2)$ iterations to achieve a convergence tolerance of $\epsilon > 0$. In each iteration, $\ell_1$LD-CTGR has a computational complexity of $\mathcal{O}(L \cdot M)$, where $L$ and $M$ denote the sample size and the number of trainable parameters, respectively. The closed-form integral computation of $\ell_1$LD-CTGR requires only $\mathcal{O}(1)$ complexity for the ultra-low-dimensional embedding with $\mathcal{D} = 2$. Hence the entire computational complexity of $\ell_1$LD-CTGR is $\mathcal{O}(L \cdot M/\epsilon^2)$, which is the same as that of GRASSP. Table A1 also shows that both GRASSP and $\ell_1$LD-CTGR have the same order of actual runtime. As for $\mathcal{D} > 2$, GRASSP requires $\mathcal{O}(\mathcal{D})$ for one integral computation in (13), based on the $\mathcal{D}$-dimensional squared $\ell_2$ distance computation. $\ell_1$LD-CTGR needs to implement a sorting of $\mathcal{D}$ zero points, which requires $\mathcal{O}(\mathcal{D}\log_2(\mathcal{D}))$ complexity. Then it needs to compute at most $(\mathcal{D} + 1)$ one-dimensional closed-form integral (with complexity $\mathcal{O}(1)$), based on (43) and (44). Hence the overall complexity of $\ell_1$LD-CTGR for one integral computation is $\mathcal{O}(\mathcal{D}\log_2(\mathcal{D})) = \tilde{\mathcal{O}}(\mathcal{D})$, which is the same asymptotic complexity as that of GRASSP. Moreover, the comparison operation is generally faster than multiplication, thus the gap between $\ell_1$LD-CTGR and GRASSP is small when $\mathcal{D}$ is small. Additional runtime experiments regarding higher dimensionalities $\mathcal{D} = 10$ and $\mathcal{D} = 100$ are provided in Table A2, which shows that $\ell_1$LD-CTGR is also efficient in these cases.

## 4 EXPERIMENTAL RESULTS

Our experimental setup extends the standard benchmark for CTGR methods outlined in (Çelikkanat et al., 2024) to include more data sets and competitors. It covers a broad range of networks with varying sizes or characteristics. Detailed experimental settings are provided in Appendix A.5.1. Eleven synthetic and real-world benchmark data sets with diverse characteristics are used for evaluation: Synthetic-$\alpha$ (Çelikkanat et al., 2022), Synthetic-$\beta$, Contacts, (Génois et al., 2015), HyperText (Isella et al., 2011), Infectious (Isella et al., 2011), Facebook (Viswanath et al., 2009), NeurIPS (Globerson et al., 2004), US Legis. (Poursafaei et al., 2022), Can. Parl. (Poursafaei et al., 2022), Wikipedia (Poursafaei et al., 2022), and Reddit (Poursafaei et al., 2022), detailed in Appendix A.5.2. NeurIPS has about 5 thousand nodes and 1.2 million edges, while Facebook has about 82 thousand nodes and 200 million edges. Both data sets contain large-scale real-world networks to test the scalability of different competitors. Eight state-of-the art methods are taken into comparisons: Node2Vec (Grover & Leskovec, 2016), CTDNE (Nguyen et al., 2018), HTNE (Zuo et al., 2018), PIVEM (Çelikkanat et al., 2022), TCL (Wang et al., 2021; Yu et al., 2023), GraphMixer (Cong et al., 2023; Yu et al., 2023), DyGFormer (Yu et al., 2023), and GRASSP (Çelikkanat et al., 2024). All these competitors have their own default settings (e.g., the embedding dimensionality), which are all adopted in our experiments. Details are provided in Appendix A.5.3. The code is available at https://github.com/laizhr/L1LD-CTGR.

The assessment is conducted in three main aspects: in-sample, out-of-sample, and across-sample evaluations. The Adam optimizer (Kingma & Ba, 2015) with a learning rate of 0.1 is used in model learning. Hyperparameters are optimized based on the best performance on the validation set.

**Network Reconstruction (in-sample)**. This evaluation focuses on reconstructing both the connection and non-connection periods within the training set, thus serving as an in-sample assessment. Its purpose is to evaluate how well a method captures the temporal structural changes occurring within the network. As shown in Table A5, $\ell_1$LD-CTGR achieves competitive performance, particularly on Synthetic-$\alpha$, HyperText, and Reddit. This indicates that $\ell_1$LD-CTGR is effective in real-world network reconstruction tasks. DyGFormer ranks the second place in this scenario, which shows its advantage in performing in-sample tasks as a Transformer-based (Vaswani et al., 2017) model.

**Network Completion (out-of-sample)**. In this evaluation, each method generalizes the learned connection structure from the training set to the test set, making it a fully out-of-sample assessment. The goal is to assess how well the method can complete the entire network with only partial information. Table 1 shows that $\ell_1$LD-CTGR outperforms other methods in all but two situations. Moreover, on the challenging real-world data sets Facebook and NeurIPS, $\ell_1$LD-CTGR maintains robust performance. Hence $\ell_1$LD-CTGR has a strong generalization capability in continuous-time network completion tasks.

Table 1: Performance of different methods for network completion (out-of-sample) across diverse data sets (mean±STD).

| Data Set | | Node2Vec | CTDNE | HTNE | PIVEM | TCL | GraphMixer | DyGFormer | GRASSP | $\ell_1$LD-CTGR |
|---|---|---|---|---|---|---|---|---|---|---|
| Synthetic-$\alpha$ | ROC | 0.459 ±0.009 | 0.489 ±0.021 | 0.542 ±0.019 | 0.578 ±0.030 | 0.602 ±0.020 | 0.619 ±0.016 | 0.680 ±0.021 | 0.559 ±0.024 | **0.817 ±0.030** |
| | PR | 0.471 ±0.013 | 0.493 ±0.019 | 0.574 ±0.020 | 0.562 ±0.027 | 0.590 ±0.016 | 0.586 ±0.011 | 0.607 ±0.020 | 0.527 ±0.021 | **0.813 ±0.031** |
| Synthetic-$\beta$ | ROC | 0.656 ±0.007 | 0.507 ±0.009 | 0.377 ±0.009 | 0.542 ±0.007 | 0.550 ±0.011 | 0.564 ±0.043 | 0.604 ±0.018 | 0.612 ±0.018 | **0.661 ±0.013** |
| | PR | **0.694 ±0.007** | 0.569 ±0.011 | 0.578 ±0.004 | 0.566 ±0.009 | 0.556 ±0.010 | 0.563 ±0.043 | 0.649 ±0.022 | 0.540 ±0.024 | 0.641 ±0.018 |
| Contacts | ROC | 0.517 ±0.021 | 0.489 ±0.029 | 0.461 ±0.025 | 0.557 ±0.009 | 0.610 ±0.001 | 0.621 ±0.002 | 0.667 ±0.016 | 0.670 ±0.016 | **0.680 ±0.017** |
| | PR | 0.526 ±0.019 | 0.553 ±0.031 | 0.509 ±0.023 | 0.579 ±0.017 | 0.602 ±0.003 | 0.687 ±0.001 | 0.711 ±0.021 | 0.714 ±0.025 | **0.724 ±0.028** |
| HyperText | ROC | 0.570 ±0.011 | 0.498 ±0.015 | 0.613 ±0.014 | 0.554 ±0.015 | 0.641 ±0.016 | 0.658 ±0.001 | 0.660 ±0.017 | 0.619 ±0.011 | **0.671 ±0.012** |
| | PR | 0.595 ±0.013 | 0.554 ±0.017 | 0.651 ±0.008 | 0.571 ±0.008 | 0.645 ±0.001 | 0.652 ±0.001 | 0.660 ±0.018 | 0.591 ±0.024 | **0.672 ±0.015** |
| Infectious | ROC | 0.681 ±0.004 | 0.534 ±0.009 | 0.651 ±0.018 | 0.578 ±0.003 | 0.728 ±0.000 | 0.724 ±0.001 | 0.726 ±0.019 | 0.728 ±0.029 | **0.756 ±0.017** |
| | PR | 0.632 ±0.011 | 0.585 ±0.008 | 0.611 ±0.016 | 0.592 ±0.004 | 0.731 ±0.001 | 0.723 ±0.003 | 0.744 ±0.018 | 0.711 ±0.028 | **0.779 ±0.017** |
| Facebook | ROC | 0.529 ±0.002 | 0.340 ±0.005 | 0.463 ±0.003 | 0.482 ±0.002 | 0.533 ±0.002 | 0.571 ±0.004 | 0.569 ±0.010 | 0.500 ±0.000 | **0.572 ±0.004** |
| | PR | 0.572 ±0.004 | 0.501 ±0.005 | 0.511 ±0.003 | 0.608 ±0.003 | 0.549 ±0.001 | 0.620 ±0.002 | 0.624 ±0.009 | 0.500 ±0.000 | **0.687 ±0.004** |
| NeurIPS | ROC | 0.355 ±0.002 | 0.455 ±0.018 | 0.222 ±0.026 | 0.469 ±0.014 | 0.503 ±0.000 | 0.467 ±0.001 | 0.496 ±0.022 | 0.360 ±0.031 | **0.533 ±0.022** |
| | PR | 0.355 ±0.002 | 0.435 ±0.022 | 0.289 ±0.028 | 0.468 ±0.027 | 0.504 ±0.000 | 0.536 ±0.002 | 0.530 ±0.018 | 0.468 ±0.026 | **0.559 ±0.019** |
| US Legis. | ROC | 0.393 ±0.003 | 0.490 ±0.009 | 0.492 ±0.014 | 0.510 ±0.010 | 0.749 ±0.006 | 0.770 ±0.015 | 0.727 ±0.016 | 0.656 ±0.013 | **0.776 ±0.013** |
| | PR | 0.486 ±0.004 | 0.534 ±0.014 | 0.542 ±0.016 | 0.529 ±0.011 | 0.684 ±0.005 | 0.707 ±0.013 | 0.699 ±0.017 | 0.587 ±0.015 | **0.725 ±0.012** |
| Can. Parl. | ROC | 0.675 ±0.003 | 0.509 ±0.010 | 0.473 ±0.011 | 0.529 ±0.012 | 0.734 ±0.008 | 0.801 ±0.014 | 0.807 ±0.010 | 0.678 ±0.009 | **0.810 ±0.009** |
| | PR | 0.616 ±0.004 | 0.568 ±0.013 | 0.538 ±0.016 | 0.545 ±0.010 | 0.692 ±0.002 | 0.739 ±0.012 | **0.874 ±0.011** | 0.709 ±0.008 | 0.761 ±0.010 |
| Wikipedia | ROC | 0.345 ±0.008 | 0.488 ±0.010 | 0.520 ±0.009 | 0.615 ±0.012 | 0.958 ±0.005 | 0.969 ±0.007 | 0.972 ±0.009 | 0.874 ±0.006 | **0.982 ±0.004** |
| | PR | 0.384 ±0.007 | 0.512 ±0.012 | 0.555 ±0.010 | 0.615 ±0.009 | 0.941 ±0.006 | 0.971 ±0.008 | 0.960 ±0.010 | 0.907 ±0.007 | **0.983 ±0.005** |
| Reddit | ROC | 0.445 ±0.010 | 0.576 ±0.012 | 0.522 ±0.011 | 0.698 ±0.013 | 0.964 ±0.008 | 0.963 ±0.008 | 0.977 ±0.006 | 0.918 ±0.009 | **0.988 ±0.005** |
| | PR | 0.471 ±0.011 | 0.531 ±0.013 | 0.580 ±0.012 | 0.632 ±0.014 | 0.965 ±0.008 | 0.963 ±0.008 | 0.980 ±0.006 | 0.910 ±0.010 | **0.983 ±0.005** |

**Future Connection Prediction (across-sample).** This evaluation involves using the sequential survival processes learned from the training set to predict future connection states in the prediction set, representing an across-sample assessment. As shown in Table 2, $\ell_1$LD-CTGR outperforms other competitors in most situations, including the challenging real-world data sets Facebook and NeurIPS. Additionally, the performance of $\ell_1$LD-CTGR is more robust than that of the other methods across different situations. Hence $\ell_1$LD-CTGR is also effective in future connection prediction tasks, even when the prediction period extends beyond the training set.

To summarize, $\ell_1$LD-CTGR is effective and robust across network reconstruction, completion, and prediction tasks, and shows a good generalization ability in the latter two tasks.

## 5 CONCLUSION AND DISCUSSION

The sequential survival process in a latent space with the squared $\ell_2$ distance is a widely-used method for continuous-time graph representation. However, the squared $\ell_2$ distance is not a valid distance for the latent metric space, because it violates the triangle inequality. The corresponding hazard function building on the squared $\ell_2$ distance may distort the relative node positions in the latent space and thus deteriorates in social, contact, and collaboration networks. Neither can the $\ell_p$ distance with $p > 1$ be

Table 2: Performance of different methods for network prediction (across-sample) across diverse data sets (mean±STD).

| Data Set | | Node2Vec | CTDNE | HTNE | PIVEM | TCL | GraphMixer | DyGFormer | GRASSP | $\ell_1$LD-CTGR |
|---|---|---|---|---|---|---|---|---|---|---|
| Synthetic-$\alpha$ | ROC | 0.486 | 0.511 | 0.575 | 0.588 | 0.420 | 0.446 | 0.427 | 0.875 | **0.922** |
| | | ±0.003 | ±0.019 | ±0.016 | ±0.014 | ±0.015 | ±0.066 | ±0.010 | ±0.020 | **±0.018** |
| | PR | 0.491 | 0.495 | 0.614 | 0.502 | 0.454 | 0.510 | 0.562 | 0.819 | **0.890** |
| | | ±0.012 | ±0.019 | ±0.020 | ±0.017 | ±0.011 | ±0.014 | ±0.012 | ±0.019 | **±0.019** |
| Synthetic-$\beta$ | ROC | 0.514 | 0.491 | 0.593 | 0.588 | 0.456 | 0.363 | 0.555 | 0.861 | **0.864** |
| | | ±0.003 | ±0.012 | ±0.006 | ±0.006 | ±0.008 | ±0.056 | ±0.009 | ±0.014 | **±0.014** |
| | PR | 0.578 | 0.555 | 0.639 | 0.598 | 0.503 | 0.465 | 0.521 | 0.829 | **0.831** |
| | | ±0.007 | ±0.018 | ±0.005 | ±0.006 | ±0.009 | ±0.035 | ±0.011 | ±0.014 | **±0.016** |
| Contacts | ROC | 0.738 | 0.509 | 0.604 | 0.493 | 0.681 | 0.676 | 0.702 | 0.763 | **0.767** |
| | | ±0.009 | ±0.016 | ±0.003 | ±0.011 | ±0.013 | ±0.004 | ±0.012 | ±0.016 | **±0.018** |
| | PR | 0.687 | 0.565 | 0.601 | 0.497 | 0.691 | 0.692 | 0.690 | 0.714 | **0.721** |
| | | ±0.015 | ±0.017 | ±0.004 | ±0.010 | ±0.003 | ±0.001 | ±0.014 | ±0.020 | **±0.018** |
| HyperText | ROC | 0.552 | 0.491 | 0.501 | 0.516 | 0.513 | 0.525 | 0.568 | **0.607** | 0.568 |
| | | ±0.003 | ±0.011 | ±0.019 | ±0.006 | ±0.005 | ±0.001 | ±0.011 | **±0.007** | ±0.005 |
| | PR | 0.518 | 0.552 | 0.502 | 0.516 | 0.525 | 0.537 | 0.556 | 0.569 | **0.576** |
| | | ±0.011 | ±0.005 | ±0.018 | ±0.004 | ±0.008 | ±0.004 | ±0.012 | ±0.009 | **±0.009** |
| Infectious | ROC | 0.869 | 0.508 | 0.730 | 0.517 | 0.867 | 0.859 | 0.887 | 0.898 | **0.901** |
| | | ±0.002 | ±0.006 | ±0.017 | ±0.008 | ±0.003 | ±0.003 | ±0.006 | ±0.015 | **±0.016** |
| | PR | 0.875 | 0.555 | 0.771 | 0.602 | 0.866 | 0.852 | 0.859 | 0.861 | **0.888** |
| | | ±0.007 | ±0.014 | ±0.013 | ±0.009 | ±0.007 | ±0.005 | ±0.007 | ±0.017 | **±0.016** |
| Facebook | ROC | 0.489 | 0.503 | 0.468 | 0.483 | 0.493 | 0.472 | 0.502 | 0.491 | **0.528** |
| | | ±0.002 | ±0.005 | ±0.003 | ±0.002 | ±0.001 | ±0.004 | ±0.008 | ±0.006 | **±0.004** |
| | PR | 0.513 | 0.517 | 0.462 | 0.491 | 0.512 | 0.517 | 0.520 | 0.498 | **0.535** |
| | | ±0.006 | ±0.005 | ±0.009 | ±0.003 | ±0.001 | ±0.002 | ±0.009 | ±0.006 | **±0.003** |
| NeurIPS | ROC | 0.445 | 0.504 | 0.510 | 0.507 | 0.5 | 0.5 | 0.502 | 0.761 | **0.778** |
| | | ±0.004 | ±0.009 | ±0.018 | ±0.014 | ±0.000 | ±0.000 | ±0.015 | ±0.010 | **±0.011** |
| | PR | 0.470 | 0.569 | 0.517 | 0.505 | 0.5 | 0.5 | 0.502 | 0.675 | **0.723** |
| | | ±0.004 | ±0.011 | ±0.022 | ±0.012 | ±0.000 | ±0.000 | ±0.014 | ±0.019 | **±0.013** |
| US Legis. | ROC | 0.475 | 0.466 | 0.490 | 0.463 | 0.482 | 0.469 | 0.742 | 0.565 | **0.754** |
| | | ±0.003 | ±0.011 | ±0.017 | ±0.012 | ±0.011 | ±0.015 | ±0.010 | ±0.012 | **±0.014** |
| | PR | 0.496 | 0.513 | 0.593 | 0.481 | 0.505 | 0.505 | 0.706 | 0.537 | **0.711** |
| | | ±0.004 | ±0.013 | ±0.020 | ±0.012 | ±0.008 | ±0.013 | ±0.011 | ±0.018 | **±0.012** |
| Can. Parl. | ROC | 0.675 | 0.509 | 0.473 | 0.529 | 0.734 | 0.801 | **0.862** | 0.678 | 0.810 |
| | | ±0.003 | ±0.010 | ±0.011 | ±0.012 | ±0.008 | ±0.014 | **±0.009** | ±0.009 | ±0.009 |
| | PR | 0.616 | 0.568 | 0.538 | 0.545 | 0.692 | 0.739 | **0.856** | 0.709 | 0.761 |
| | | ±0.004 | ±0.013 | ±0.016 | ±0.010 | ±0.002 | ±0.012 | **±0.010** | ±0.008 | ±0.010 |
| Wikipedia | ROC | 0.393 | 0.415 | 0.445 | 0.502 | 0.823 | 0.841 | 0.729 | 0.655 | **0.845** |
| | | ±0.010 | ±0.012 | ±0.011 | ±0.009 | ±0.007 | ±0.006 | ±0.010 | ±0.008 | **±0.007** |
| | PR | 0.398 | 0.374 | 0.425 | 0.489 | 0.860 | **0.884** | 0.758 | 0.654 | 0.836 |
| | | ±0.011 | ±0.013 | ±0.012 | ±0.010 | ±0.008 | **±0.007** | ±0.009 | ±0.008 | ±0.007 |
| Reddit | ROC | 0.473 | 0.503 | 0.524 | 0.517 | 0.840 | 0.829 | 0.863 | 0.768 | **0.882** |
| | | ±0.011 | ±0.012 | ±0.013 | ±0.014 | ±0.009 | ±0.010 | ±0.011 | ±0.011 | **±0.008** |
| | PR | 0.454 | 0.460 | 0.490 | 0.486 | 0.877 | 0.852 | **0.904** | 0.751 | 0.895 |
| | | ±0.010 | ±0.012 | ±0.013 | ±0.014 | ±0.009 | ±0.010 | **±0.011** | ±0.011 | ±0.008 |

used, because the corresponding integral computation is intractable. To address these problems, we develop a theoretically-sound $\ell_1$ latent distance based continuous-time graph representation ($\ell_1$LD-CTGR). Specifically, we verify that the average $\ell_1$ distance during an event time enjoys a similar bounded property to that of the squared $\ell_2$ distance case. We also derive the integral of the hazard function for the $\ell_1$ distance case, which results in a closed-form piece-wise exponential integral. This differs markedly from the squared $\ell_2$ distance case, where the corresponding integral is the well-known Gaussian integral. We further develop an efficient integral computing algorithm for the ultra-low-dimensional embedding. Finally, although the $\ell_1$ distance, hazard function, and survival function are non-differentiable, we successfully find a descent direction for the hazard function to replace the gradient. This allows us to leverage mainstream learning architectures such as Pytorch to optimize the model parameters.

Extensive experiments conducted on eleven synthetic and real-world benchmark data sets show that $\ell_1$LD-CTGR achieves good performance across network reconstruction, completion, and prediction tasks, outperforming eight state-of-the-art methods in most cases. Moreover, $\ell_1$LD-CTGR not only achieves the same computational efficiency as GRASSP, but also facilitates a true latent metric space. Future works may focus on exploring additional types of valid metrics to further improve the geometric diversity of the latent space.

ACKNOWLEDGMENTS

This work is supported in part by the National Natural Science Foundation of China (grant numbers: 62541606, 62176103, 62276114, and 62206110), in part by the National Research Foundation, Singapore and Infocomm Media Development Authority under its Trust Tech Funding Initiative, and in part by The Major Key Project of PCL (No. PCL2025A02 and No. PCL2024A04). Any opinions, findings and conclusions or recommendations expressed in this material are those of the author(s) and do not reflect the views of National Research Foundation, Singapore and Infocomm Media Development Authority.

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

# A  APPENDIX

## A.1  PROOF OF PROPOSITION 1

To prove this proposition, we need the following Jensen's inequality.

**Lemma 1** (Jensen's inequality). *Let $f : [a, b] \to \mathbb{R}^+$ be a non-negative Lebesgue-integrable function, and $\varphi : \mathbb{R} \to \mathbb{R}$ be a convex Lebesgue-integrable function. Then:*

$$\frac{1}{b-a} \int_a^b \varphi\left(f(t)\right) \, \mathrm{d}t \geqslant \varphi\left(\frac{1}{b-a} \int_a^b f(t) \, \mathrm{d}t\right). \tag{25}$$

*Proof.* We begin by examining the integral $\frac{1}{e_{k+1}-e_k} \int_{e_k}^{e_{k+1}} \lambda_{ij}(s,t) \, \mathrm{d}t$. It can be seen as the average value of $\lambda_{ij}(s,t)$ in the interval $[e_k, e_{k+1})$. It can control the average $\ell_1$ distance by using convexity and Jensen's inequality.

$$\frac{1}{e_{k+1}-e_k} \int_{e_k}^{e_{k+1}} \exp\left(\beta(s) + s\|\boldsymbol{r}_i(t) - \boldsymbol{r}_j(t)\|_1\right) \, \mathrm{d}t$$

$$= \exp(\beta(s)) \left[\frac{1}{e_{k+1}-e_k} \int_{e_k}^{e_{k+1}} \exp\left(s\|\boldsymbol{r}_i(t) - \boldsymbol{r}_j(t)\|_1\right) \, \mathrm{d}t\right]$$

$$\geqslant \exp(\beta(s)) \exp\left(\frac{s}{e_{k+1}-e_k} \int_{e_k}^{e_{k+1}} \|\boldsymbol{r}_i(t) - \boldsymbol{r}_j(t)\|_1 \, \mathrm{d}t\right). \tag{26}$$

The above inequality follows from Lemma 1 with $f(t) := \|\boldsymbol{r}_i(t) - \boldsymbol{r}_j(t)\|_1$, $\varphi(x) := \exp(s \cdot x)$, $a := e_k$ and $b := e_{k+1}$. Note that $\varphi(x)$ is convex w.r.t. $x$ for both $s = 1$ and $s = -1$. Taking the logarithm of both sides of (26) yields:

$$\log\left(\int_{e_k}^{e_{k+1}} \lambda_{ij}(s,t) \, \mathrm{d}t\right) - \log(e_{k+1} - e_k)$$

$$\geqslant \beta(s) + \frac{s}{e_{k+1}-e_k} \int_{e_k}^{e_{k+1}} \|\boldsymbol{r}_i(t) - \boldsymbol{r}_j(t)\|_1 \, \mathrm{d}t. \tag{27}$$

On the other hand, it follows from the definition of the survival function (1) that

$$\int_{e_k}^{e_{k+1}} \lambda_{ij}(s,t) \, \mathrm{d}t = \log(S_{ij}(e_k)) - \log(S_{ij}(e_{k+1})). \tag{28}$$

Inserting (28) into (27) and rearranging terms yield:

$$\frac{s}{e_{k+1}-e_k} \int_{e_k}^{e_{k+1}} \|\boldsymbol{r}_i(t) - \boldsymbol{r}_j(t)\|_1 \, \mathrm{d}t$$

$$\leqslant \log\left(\log(S_{ij}(e_k)) - \log(S_{ij}(e_{k+1}))\right) - \beta(s) - \log(e_{k+1} - e_k). \tag{29}$$

By setting $s = 1$ and $s = -1$ in (29), we have

$$\frac{1}{e_{k+1}-e_k} \int_{e_k}^{e_{k+1}} \|\boldsymbol{r}_i(t) - \boldsymbol{r}_j(t)\|_1 \, \mathrm{d}t$$

$$\leqslant \log\left(\log(S_{ij}(e_k)) - \log(S_{ij}(e_{k+1}))\right) - \beta(1) - \log(e_{k+1} - e_k), \tag{30}$$

$$\frac{-1}{e_{k+1}-e_k} \int_{e_k}^{e_{k+1}} \|\boldsymbol{r}_i(t) - \boldsymbol{r}_j(t)\|_1 \, \mathrm{d}t$$

$$\leqslant \log\left(\log(S_{ij}(e_k)) - \log(S_{ij}(e_{k+1}))\right) - \beta(-1) - \log(e_{k+1} - e_k), \tag{31}$$

Multiplying both sides of (31) by $s = -1$, then the upper and lower bounds in (15) can be derived. $\square$

## A.2  PROOF OF THEOREM 1

*Proof.* We focus on the latter part of (16):

$$
\int_0^{e_u-e_l} \exp\left(s\|\Delta\boldsymbol{x}_{ij}+\Delta\boldsymbol{v}_{ij}t\|_1\right)\,\mathrm{d}t
$$

$$
=\int_0^{e_u-e_l} \exp\left(s\sum_{d=1}^{\mathcal{D}}|\Delta x_{ij,d}+\Delta v_{ij,d}t|\right)\,\mathrm{d}t, \tag{32}
$$

where $\Delta x_{ij,d}$ denotes the $d$-th dimension of $\Delta\boldsymbol{x}_{ij}$ and $\mathcal{D}$ is the dimensionality of the latent space.

The key technique is to handle the $\mathcal{D}$ absolute operators simultaneously within the integral interval $[0, e_u - e_l)$. First, since $|\Delta x_{ij,d}+\Delta v_{ij,d}t| = |-\Delta x_{ij,d}-\Delta v_{ij,d}t|$, we can adjust the sign of $\Delta v_{ij,d}$ to ensure that it is non-negative:

$$
\Delta v_{ij,d}\leftarrow -\Delta v_{ij,d},\ \Delta x_{ij,d}\leftarrow -\Delta x_{ij,d}, \quad \text{if } \Delta v_{ij,d} < 0. \tag{33}
$$

Next, we find the zero point $z_{ij,d}$ for the equation $\Delta x_{ij,d} + \Delta v_{ij,d}t = 0$:

$$
\begin{cases} z_{ij,d}:= -\dfrac{\Delta x_{ij,d}}{\Delta v_{ij,d}} & \text{if } \Delta v_{ij,d}\neq 0 \\ z_{ij,d}:= 0,\ \Delta x_{ij,d}\leftarrow|\Delta x_{ij,d}| & \text{if } \Delta v_{ij,d}=0 \end{cases}, \quad d=1,2,\cdots,\mathcal{D}. \tag{34}
$$

When $\Delta v_{ij,d} = 0$, we can set $z_{ij,d} = 0$ and assign $|\Delta x_{ij,d}|$ to $\Delta x_{ij,d}$. Then we proceed with the following steps. We sort all the zero points in the ascending order:

$$
z_{ij,(1)} \leqslant z_{ij,(2)} \leqslant \cdots \leqslant z_{ij,(\mathcal{D})}. \tag{35}
$$

Accordingly, we denote $\Delta x_{ij,(d)}$ and $\Delta v_{ij,(d)}$ as having the same sorting order $(d)$ as that of $z_{ij,(d)}$. Then the sum in (32) over the interval $[z_{ij,(c)}, z_{ij,(c+1)})$ becomes

$$
\sum_{d=1}^{\mathcal{D}}|\Delta x_{ij,d}+\Delta v_{ij,d}t|
$$

$$
=\sum_{d=1}^{\mathcal{D}}|\Delta x_{ij,(d)}+\Delta v_{ij,(d)}t|
$$

$$
=\sum_{d=1}^{c}(\Delta v_{ij,(d)}t+\Delta x_{ij,(d)}) + \sum_{\tilde{d}=c+1}^{\mathcal{D}}(-\Delta x_{ij,(\tilde{d})}-\Delta v_{ij,(\tilde{d})}t)
$$

$$
=\underbrace{\left(\sum_{d=1}^{c}\Delta v_{ij,(d)}-\sum_{\tilde{d}=c+1}^{\mathcal{D}}\Delta v_{ij,(\tilde{d})}\right)}t+\underbrace{\underbrace{\left(\sum_{d=1}^{c}\Delta x_{ij,(d)}-\sum_{\tilde{d}=c+1}^{\mathcal{D}}\Delta x_{ij,(\tilde{d})}\right)}}
$$

$$
=: \underline{V_{ij,(c)}}t+\underline{\underline{X_{ij,(c)}}}, \tag{36}
$$

where $V_{ij,(c)}$ and $X_{ij,(c)}$ denote the linear coefficient of $t$ and the constant term on the interval $[z_{ij,(c)}, z_{ij,(c+1)})$, respectively. Therefore, if $V_{ij,(c)} \neq 0$, the part of the integral (32) over the interval $[z_{ij,(c)}, z_{ij,(c+1)})$ can be computed by

$$
\mathcal{I}_{ij,(c),(c+1)} := \int_{z_{ij,(c)}}^{z_{ij,(c+1)}} \exp\left(s\sum_{d=1}^{\mathcal{D}}|\Delta x_{ij,d}+\Delta v_{ij,d}t|\right)\,\mathrm{d}t
$$

$$
=\int_{z_{ij,(c)}}^{z_{ij,(c+1)}} \exp\left(s\left(V_{ij,(c)}t+X_{ij,(c)}\right)\right)\,\mathrm{d}t
$$

$$
=\exp\left(sX_{ij,(c)}\right)\cdot\frac{1}{sV_{ij,(c)}}\exp\left(sV_{ij,(c)}t\right)|_{z_{ij,(c)}}^{z_{ij,(c+1)}}
$$

$$
=\frac{\exp\left(sX_{ij,(c)}\right)}{sV_{ij,(c)}}\left(\exp\left(sV_{ij,(c)}z_{ij,(c+1)}\right)-\exp\left(sV_{ij,(c)}z_{ij,(c)}\right)\right). \tag{37}
$$

If $V_{ij,(c)} = 0$, then

$$
\begin{aligned}
\mathcal{I}_{ij,(c),(c+1)} &:= \int_{z_{ij,(c)}}^{z_{ij,(c+1)}} \exp\left(s\left(V_{ij,(c)}t + X_{ij,(c)}\right)\right)\,\mathrm{d}t \\
&= \exp\left(sX_{ij,(c)}\right)\left(z_{ij,(c+1)} - z_{ij,(c)}\right).
\end{aligned}
\tag{38}
$$

Next, we need to further investigate the case of $z_{ij,(1)} > 0$ or $z_{ij,(\mathcal{D})} < e_u - e_l$, where the lower or upper limit of the integral goes beyond the smallest or largest zero point. In both cases, we can extend the definitions of $V_{ij,(c)}$ and $X_{ij,(c)}$ to $c = 0$ and $c = \mathcal{D}$:

$$
V_{ij,(0)} := -\sum_{\tilde{d}=1}^{\mathcal{D}} \Delta v_{ij,(\tilde{d})}, \quad X_{ij,(0)} := -\sum_{\tilde{d}=1}^{\mathcal{D}} \Delta x_{ij,(\tilde{d})},
$$

$$
V_{ij,(\mathcal{D})} := \sum_{d=1}^{\mathcal{D}} \Delta v_{ij,(d)}, \quad X_{ij,(\mathcal{D})} := \sum_{d=1}^{\mathcal{D}} \Delta x_{ij,(d)}.
\tag{39}
$$

Let $z_{ij,(\underline{c})}$ be the smallest zero point that is greater than 0, and $z_{ij,(\overline{c})}$ be the largest zero point that is less than $(e_u - e_l)$. We compute the following two integrals:

$$
\begin{aligned}
\mathcal{I}_{ij,0,(\underline{c})} &:= \int_0^{z_{ij,(\underline{c})}} \exp\left(s\left(V_{ij,(\underline{c}-1)}t + X_{ij,(\underline{c}-1)}\right)\right)\,\mathrm{d}t \\
&= \begin{cases} \frac{\exp\left(sX_{ij,(\underline{c}-1)}\right)}{sV_{ij,(\underline{c}-1)}}\left(\exp\left(sV_{ij,(\underline{c}-1)}z_{ij,(\underline{c})}\right) - 1\right) & \text{if } V_{ij,(\underline{c}-1)} \neq 0 \\ \exp\left(sX_{ij,(\underline{c}-1)}\right)\cdot z_{ij,(\underline{c})} & \text{if } V_{ij,(\underline{c}-1)} = 0 \end{cases},
\end{aligned}
\tag{40}
$$

$$
\begin{aligned}
\mathcal{I}_{ij,(\overline{c}),e_u-e_l} &:= \int_{z_{ij,(\overline{c})}}^{e_u-e_l} \exp\left(s\left(V_{ij,(\overline{c})}t + X_{ij,(\overline{c})}\right)\right)\,\mathrm{d}t \\
&= \begin{cases} \frac{\exp\left(sX_{ij,(\overline{c})}\right)}{sV_{ij,(\overline{c})}}\left(\exp\left(sV_{ij,(\overline{c})}(e_u - e_l)\right) - \exp\left(sV_{ij,(\overline{c})}z_{ij,(\overline{c})}\right)\right) & \text{if } V_{ij,(\overline{c})} \neq 0 \\ \exp\left(sX_{ij,(\overline{c})}\right)\cdot(e_u - e_l - z_{ij,(\overline{c})}) & \text{if } V_{ij,(\overline{c})} = 0 \end{cases}.
\end{aligned}
\tag{41}
$$

If $z_{ij,(\underline{c})} > e_u - e_l$ or $z_{ij,(\overline{c})} < 0$, then the integrand will not change sign in the interval $[0, e_u - e_l)$. In either case, we denote $(\tilde{c}) := (\underline{c} - 1)$ or $(\tilde{c}) := (\overline{c})$, respectively. Then the integral (32) has the following simplest form:

$$
\begin{aligned}
\mathcal{I}_{ij,0,e_u-e_l} &:= \int_0^{e_u-e_l} \exp\left(s\left(V_{ij,(\tilde{c})}t + X_{ij,(\tilde{c})}\right)\right)\,\mathrm{d}t \\
&= \begin{cases} \frac{\exp\left(sX_{ij,(\tilde{c})}\right)}{sV_{ij,(\tilde{c})}}\left(\exp\left(sV_{ij,(\tilde{c})}(e_u - e_l)\right) - 1\right) & \text{if } V_{ij,(\tilde{c})} \neq 0 \\ \exp\left(sX_{ij,(\tilde{c})}\right)\cdot(e_u - e_l) & \text{if } V_{ij,(\tilde{c})} = 0 \end{cases}.
\end{aligned}
\tag{42}
$$

In other cases, the integral (32) can be computed by:

$$
\begin{aligned}
&\int_0^{e_u-e_l} \exp\left(s\sum_{d=1}^{\mathcal{D}} |\Delta x_{ij,d} + \Delta v_{ij,d}t|\right)\,\mathrm{d}t \\
&= \mathcal{I}_{ij,0,(\underline{c})} + \sum_{c=\underline{c}}^{\overline{c}-1} \mathcal{I}_{ij,(c),(c+1)} + \mathcal{I}_{ij,(\overline{c}),e_u-e_l},
\end{aligned}
\tag{43}
$$

with $\mathcal{I}_{ij,(c),(c+1)}, \mathcal{I}_{ij,0,(\underline{c})}$ and $\mathcal{I}_{ij,(\overline{c}),e_u-e_l}$ defined in (37), (38), (40), and (41). Summarizing all the analyses of this proof, Theorem 1 indicates that the integral of the hazard function based on the $\ell_1$ distance is a closed-form piece-wise exponential integral. $\square$

**Note:** When computing $\sum_{c=\underline{c}}^{\overline{c}-1} \mathcal{I}_{ij,(c),(c+1)}$ in (17), it is unnecessary to compute each $V_{ij,(c)}$ or $X_{ij,(c)}$ individually as per (36). Instead, we can exploit the following relationships to compute $\mathcal{I}_{ij,(c),(c+1)}$ successively from $c = \underline{c}$ to $c = \overline{c} - 1$.

$$
V_{ij,(c+1)} = V_{ij,(c)} + 2\Delta v_{ij,(c+1)}, \quad X_{ij,(c+1)} = X_{ij,(c)} + 2\Delta x_{ij,(c+1)}.
\tag{44}
$$

**A demonstration of Theorem 1:** Let the hazard function

$$\lambda_{ij}(s,t) = \exp\left(|t - 0.1| + |t - 0.2|\right) \tag{45}$$

with $e_l = 0$ and $e_u = 1$. Its plot is shown in Figure A1. The zero points are $z_{ij,(\underline{c})} = 0.1$ and $z_{ij,(\overline{c})} = 0.2$. Therefore, the entire integral should be divided into 3 parts and computed using the second line of (17):

$$\int_0^1 \lambda_{ij}(s,t)\,\mathrm{d}t$$
$$= \int_0^1 \exp\left(|t - 0.1| + |t - 0.2|\right)\,\mathrm{d}t$$
$$= \int_0^{0.1} \exp\left(0.3 - 2t\right)\,\mathrm{d}t + \int_{0.1}^{0.2} \exp(0.1)\,\mathrm{d}t + \int_{0.2}^1 \exp\left(2t - 0.3\right)\,\mathrm{d}t$$
$$= -\frac{1}{2}\exp\left(0.3 - 2t\right)\big|_0^{0.1} + 0.1\exp(0.1) + \frac{1}{2}\exp\left(2t - 0.3\right)\big|_{0.2}^1$$
$$= 0.5(\exp(1.7) + \exp(0.3)) - 0.9\exp(0.1)$$
$$\approx 2.4172. \tag{46}$$

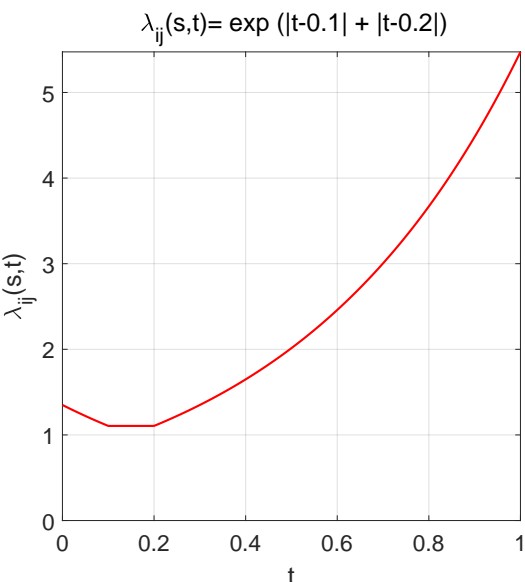

Figure A1: A simple example of $\lambda_{ij}(s,t)$ in the interval $[0,1]$.

### A.3 PROOF OF THEOREM 2

*Proof.* When $\mathcal{D} = 2$, the integral (32) becomes

$$\int_0^{e_{k+1} - e_k} \exp\left(s(|\Delta x_{ij,1} + \Delta v_{ij,1}t| + |\Delta x_{ij,2} + \Delta v_{ij,2}t|)\right)\,\mathrm{d}t. \tag{47}$$

Then we examine the core part in the exponent:

$$f(t) := |\Delta x_{ij,1} + \Delta v_{ij,1}t| + |\Delta x_{ij,2} + \Delta v_{ij,2}t|. \tag{48}$$

In CTGR, the probability of $\Delta v_{ij,d} = 0$ is essentially zero, especially when each node pair has relative movement at any dimension and any time. Hence with a slight abuse of definition, we assume $\Delta v_{ij,d} \neq 0$, $d = 1, 2$. Let $u_{ij,d} := \frac{\Delta x_{ij,d}}{\Delta v_{ij,d}}$ for $d = 1, 2$, and sort $u_{ij,d}$ such that $u_{ij,(1)} \leqslant u_{ij,(2)}$. Then (48) becomes

$$f(t) := |\Delta v_{ij,(1)}||u_{ij,(1)} + t| + |\Delta v_{ij,(2)}||u_{ij,(2)} + t|. \tag{49}$$

As the argument $t$ falls into 3 different intervals, $f(t)$ also falls into 3 different segments:

$$
\begin{cases}
f_1(t) := -(\operatorname{sign}(\Delta v_{ij,(1)})\Delta x_{ij,(1)} + \operatorname{sign}(\Delta v_{ij,(2)})\Delta x_{ij,(2)}) \\
\qquad -(|\Delta v_{ij,(1)}| + |\Delta v_{ij,(2)}|)t & \text{if } t < -u_{ij,(2)} \\
f_2(t) := (\operatorname{sign}(\Delta v_{ij,(2)})\Delta x_{ij,(2)} - \operatorname{sign}(\Delta v_{ij,(1)})\Delta x_{ij,(1)}) \\
\qquad +(|\Delta v_{ij,(2)}| - |\Delta v_{ij,(1)}|)t & \text{if } -u_{ij,(2)} \leqslant t \leqslant -u_{ij,(1)} \\
f_3(t) := (\operatorname{sign}(\Delta v_{ij,(2)})\Delta x_{ij,(2)} + \operatorname{sign}(\Delta v_{ij,(1)})\Delta x_{ij,(1)}) \\
\qquad +(|\Delta v_{ij,(2)}| + |\Delta v_{ij,(1)}|)t & \text{if } t > -u_{ij,(1)}
\end{cases}
\tag{50}
$$

where the $\operatorname{sign}$ function is defined as follows:

$$
\operatorname{sign} : \mathbb{R} \to \mathbb{R}, \quad \operatorname{sign}(x) := \begin{cases} 1 & \text{if } x > 0 \\ 0 & \text{if } x = 0 \\ -1 & \text{if } x < 0 \end{cases}.
\tag{51}
$$

Note that it can operate on a tensor in an element-wise way. Next, we need to break the integration interval $[0, e_{k+1} - e_k)$ into 3 parts according to the 3 segments in (50). Moreover, we should also consider the relative positions of the break points $-u_{ij,(1)}$, $-u_{ij,(2)}$ and the integration limits $0$, $(e_{k+1} - e_k)$. To tackle this issue, we define the following two break points:

$$
\begin{aligned}
\underline{\zeta} &:= \min\{\max\{0, -u_{ij,(2)}\}, e_{k+1} - e_k\}, \\
\overline{\zeta} &:= \min\{\max\{0, -u_{ij,(1)}\}, e_{k+1} - e_k\}.
\end{aligned}
\tag{52}
$$

Then the 3 integrals regarding the 3 segments of $f(t)$ are:

$$
\begin{aligned}
\mathcal{I}_1 &:= \int_0^{\underline{\zeta}} \exp(s f_1(t)) \, \mathrm{d}t \\
&= \underline{\zeta} \exp(-s(\operatorname{sign}(\Delta v_{ij,(1)})\Delta x_{ij,(1)} + \operatorname{sign}(\Delta v_{ij,(2)})\Delta x_{ij,(2)})) \\
&\quad + \exp(-s\underline{\zeta}(|\Delta v_{ij,(1)}| + |\Delta v_{ij,(2)}|)),
\end{aligned}
\tag{53}
$$

$$
\begin{aligned}
\mathcal{I}_2 &:= \int_{\underline{\zeta}}^{\overline{\zeta}} \exp(s f_2(t)) \, \mathrm{d}t \\
&= (\overline{\zeta} - \underline{\zeta}) \exp(s(\operatorname{sign}(\Delta v_{ij,(2)})\Delta x_{ij,(2)} - \operatorname{sign}(\Delta v_{ij,(1)})\Delta x_{ij,(1)})) \\
&\quad + \exp(s(\overline{\zeta} - \underline{\zeta})(|\Delta v_{ij,(2)}| - |\Delta v_{ij,(1)}|)),
\end{aligned}
\tag{54}
$$

$$
\begin{aligned}
\mathcal{I}_3 &:= \int_{\overline{\zeta}}^{e_{k+1} - e_k} \exp(s f_3(t)) \, \mathrm{d}t \\
&= (e_{k+1} - e_k - \overline{\zeta}) \exp(s(\operatorname{sign}(\Delta v_{ij,(2)})\Delta x_{ij,(2)} + \operatorname{sign}(\Delta v_{ij,(1)})\Delta x_{ij,(1)})) \\
&\quad + \exp(s(e_{k+1} - e_k - \overline{\zeta})(|\Delta v_{ij,(2)}| + |\Delta v_{ij,(1)}|)).
\end{aligned}
\tag{55}
$$

The entire integral is

$$
\int_{e_k}^{e_{k+1}} \lambda_{ij}(s, t) \, \mathrm{d}t = \exp(\beta(s))(\mathcal{I}_1 + \mathcal{I}_2 + \mathcal{I}_3).
\tag{56}
$$

To see how $\left\{ \int_{e_k}^{e_{k+1}} \lambda_{ij}(s, t) \, \mathrm{d}t \right\}_{i,j,k}$ can be computed in a tensor-parallelized way, we first indicate that all the operators from (48) to (56) are tensor-parallel. These operators include $\operatorname{sign}$, $\exp$, the absolute value operator, element-wise division, and others. Therefore, we can stack each variable w.r.t. all the observations (one line for each observation) into a single tensor as follows:

$$
\begin{aligned}
\Delta \boldsymbol{v}_{(d)} &:= [\Delta v_{ij,(d)}(e_k)]_{i,j,k}, \quad \Delta \boldsymbol{x}_{(d)} := [\Delta x_{ij,(d)}(e_k)]_{i,j,k}, \quad d = 1, 2, \\
\boldsymbol{s} &:= [s(i, j, e_k)]_{i,j,k}, \quad \underline{\boldsymbol{\zeta}} := [\underline{\zeta}(i, j, e_k)]_{i,j,k}, \quad \overline{\boldsymbol{\zeta}} := [\overline{\zeta}(i, j, e_k)]_{i,j,k}, \\
\boldsymbol{e}_k &:= [e_k(i, j)]_{i,j,k}, \quad \boldsymbol{e}_{k+1} := [e_{k+1}(i, j)]_{i,j,k}.
\end{aligned}
\tag{57}
$$

These are all $L \times 1$ tensors if there are $L$ observations. Accordingly, the 3 stacked (tensor-parallelized) integrals for (53),(54), and (55) can be computed by

$$
\boldsymbol{\mathcal{I}}_1 = \underline{\boldsymbol{\zeta}} \odot \exp(-\boldsymbol{s} \odot (\operatorname{sign}(\Delta \boldsymbol{v}_{(1)}) \odot \Delta \boldsymbol{x}_{(1)} + \operatorname{sign}(\Delta \boldsymbol{v}_{(2)}) \odot \Delta \boldsymbol{x}_{(2)})
$$

$$+ \exp(-\boldsymbol{s} \odot \underline{\boldsymbol{\zeta}} \odot (|\Delta \boldsymbol{v}_{(1)}| + |\Delta \boldsymbol{v}_{(2)}|)), \tag{58}$$

$$\boldsymbol{\mathcal{I}}_2 = (\overline{\boldsymbol{\zeta}} - \underline{\boldsymbol{\zeta}}) \odot \exp(\boldsymbol{s} \odot (\mathrm{sign}(\Delta \boldsymbol{v}_{(2)}) \odot \Delta \boldsymbol{x}_{(2)} - \mathrm{sign}(\Delta \boldsymbol{v}_{(1)}) \odot \Delta \boldsymbol{x}_{(1)})$$
$$+ \exp(\boldsymbol{s} \odot (\overline{\boldsymbol{\zeta}} - \underline{\boldsymbol{\zeta}}) \odot (|\Delta \boldsymbol{v}_{(2)}| - |\Delta \boldsymbol{v}_{(1)}|)), \tag{59}$$

$$\boldsymbol{\mathcal{I}}_3 = (\boldsymbol{e}_{k+1} - \boldsymbol{e}_k - \overline{\boldsymbol{\zeta}}) \odot \exp(\boldsymbol{s} \odot (\mathrm{sign}(\Delta \boldsymbol{v}_{(2)}) \odot \Delta \boldsymbol{x}_{(2)} + \mathrm{sign}(\Delta \boldsymbol{v}_{(1)}) \odot \Delta \boldsymbol{x}_{(1)})$$
$$+ \exp(\boldsymbol{s} \odot (\boldsymbol{e}_{k+1} - \boldsymbol{e}_k - \overline{\boldsymbol{\zeta}}) \odot (|\Delta \boldsymbol{v}_{(2)}| + |\Delta \boldsymbol{v}_{(1)}|)), \tag{60}$$

where $\odot$ denotes the element-wise multiplication. Finally, the entire stacked integral can be computed by (18).

$\square$

### A.4 PROOF OF THEOREM 3

*Proof.* We can omit the variable $t$ in the expression of $\lambda_{ij}$ for simplicity, which does not affect the proof. We start from the simplest case.

**Case 1: $\mathcal{C} = \mathcal{D}$.** In this case, $\lambda_{ij}(\boldsymbol{r}_i) = \exp(\beta(s))$. Hence $\partial \lambda_{ij}(\boldsymbol{r}_i) = \nabla \lambda_{ij}(\boldsymbol{r}_i) = \boldsymbol{0}$.

**Case 2: $\mathcal{C} \subsetneq \mathcal{D}$.** We need to prove that there exists some $\delta > 0$ such that $\lambda_{ij}(\boldsymbol{r}_i - \delta \partial \lambda_{ij}(\boldsymbol{r}_i)) < \lambda_{ij}(\boldsymbol{r}_i)$. Taking a subtraction yields

$$\lambda_{ij}(\boldsymbol{r}_i - \delta \partial \lambda_{ij}(\boldsymbol{r}_i)) - \lambda_{ij}(\boldsymbol{r}_i)$$
$$= \exp(\beta(s) + s\|\boldsymbol{r}_i(t) - \boldsymbol{r}_j(t)\|_1)$$
$$\cdot [\exp(s(\|\boldsymbol{r}_i - \delta \partial \lambda_{ij}(\boldsymbol{r}_i) - \boldsymbol{r}_j\|_1 - \|\boldsymbol{r}_i - \boldsymbol{r}_j\|_1)) - 1]. \tag{61}$$

Hence it suffices to prove that the second multiplier is negative. Denote $\partial \lambda_{ij,d}(\boldsymbol{r}_i)$ as the $d$-th dimension of $\partial \lambda_{ij}(\boldsymbol{r}_i)$. For all $d \in \mathcal{C}$, $r_{i,d} = r_{j,d}$ and thus $\partial \lambda_{ij,d}(\boldsymbol{r}_i) = 0$. For all $d \notin \mathcal{C}$, it can be seen that $\partial \lambda_{ij,d}(\boldsymbol{r}_i) \neq 0$. Then we can set

$$\delta := \min_{d \notin \mathcal{C}} \frac{|r_{i,d} - r_{j,d}|}{2|\partial \lambda_{ij,d}(\boldsymbol{r}_i)|}. \tag{62}$$

This is a uniform step size such that $\mathrm{sign}(r_{i,d} - \delta \partial \lambda_{ij,d}(\boldsymbol{r}_i) - r_{j,d}) = \mathrm{sign}(r_{i,d} - r_{j,d})$ for all $d \notin \mathcal{C}$. We further set

$$\overline{\delta} = \delta / \exp(\beta(s) + s\|\boldsymbol{r}_i(t) - \boldsymbol{r}_j(t)\|_1) > 0, \tag{63}$$

then

$$\exp(s(\|\boldsymbol{r}_i - \delta \partial \lambda_{ij}(\boldsymbol{r}_i) - \boldsymbol{r}_j\|_1 - \|\boldsymbol{r}_i - \boldsymbol{r}_j\|_1)) - 1$$
$$= \exp\left(s \sum_{d \notin \mathcal{C}} (|(r_{i,d} - \overline{\delta}s \cdot \mathrm{sign}(r_{i,d} - r_{j,d}) - r_{j,d}| - |r_{i,d} - r_{j,d}|)\right) - 1$$
$$= \exp\left(s \sum_{d \notin \mathcal{C}} \mathrm{sign}(r_{i,d} - r_{j,d})((r_{i,d} - \overline{\delta}s \cdot \mathrm{sign}(r_{i,d} - r_{j,d}) - r_{j,d}) - (r_{i,d} - r_{j,d}))\right) - 1$$
$$= \exp\left(s \sum_{d \notin \mathcal{C}} \mathrm{sign}(r_{i,d} - r_{j,d})(-\overline{\delta}s \cdot \mathrm{sign}(r_{i,d} - r_{j,d}))\right) - 1$$
$$= \exp\left(-\overline{\delta} \sum_{d \notin \mathcal{C}} s^2(\mathrm{sign}(r_{i,d} - r_{j,d}))^2\right) - 1$$
$$= \exp(-\overline{\delta}|\mathcal{D} \backslash \mathcal{C}|) - 1, \tag{64}$$

where $|\mathcal{D} \backslash \mathcal{C}|$ denotes the cardinality of the index set $\mathcal{D} \backslash \mathcal{C}$. The last equality holds because $s^2 = 1$ and $(\mathrm{sign}(r_{i,d} - r_{j,d}))^2 = 1$. Since $\mathcal{C} \subsetneq \mathcal{D}$, we have $|\mathcal{D} \backslash \mathcal{C}| \geqslant 1$ and thus $\exp(-\overline{\delta}|\mathcal{D} \backslash \mathcal{C}|) - 1 < 0$. From (64) back to (61), it can be seen that $\lambda_{ij}(\boldsymbol{r}_i - \delta \partial \lambda_{ij}(\boldsymbol{r}_i)) < \lambda_{ij}(\boldsymbol{r}_i)$ and $-\partial \lambda_{ij}(\boldsymbol{r}_i)$ is a descent direction of $\lambda_{ij}$ at $\boldsymbol{r}_i$.

**Case 3: $\mathcal{C} = \emptyset$.** We need to verify that the following inequality holds for any $\boldsymbol{y} \to \boldsymbol{r}_i$:

$$\lambda_{ij}(\boldsymbol{y}) - \lambda_{ij}(\boldsymbol{r}_i) \geqslant (\partial \lambda_{ij}(\boldsymbol{r}_i))^\top (\boldsymbol{y} - \boldsymbol{r}_i). \tag{65}$$

By substituting the expressions of $\lambda_{ij}(\boldsymbol{y})$, $\lambda_{ij}(\boldsymbol{r}_i)$, and $\partial\lambda_{ij}(\boldsymbol{r}_i)$, and rearranging terms, the inequality (65) can be transformed into

$$
\begin{aligned}
\exp\left(s(\|\boldsymbol{y}-\boldsymbol{r}_j\|_1-\|\boldsymbol{r}_i-\boldsymbol{r}_j\|_1)\right)-1 \\
\geqslant s\cdot\mathrm{sign}(\boldsymbol{r}_i-\boldsymbol{r}_j)^\top(\boldsymbol{y}-\boldsymbol{r}_i) \\
= s\cdot\mathrm{sign}(\boldsymbol{r}_i-\boldsymbol{r}_j)^\top((\boldsymbol{y}-\boldsymbol{r}_j)-(\boldsymbol{r}_i-\boldsymbol{r}_j)).
\end{aligned}
\tag{66}
$$

Since $\mathcal{C}=\emptyset$, we define

$$
\epsilon := \frac{\min_{d\in\mathcal{D}}|r_{i,d}-r_{j,d}|}{2} > 0.
\tag{67}
$$

Then when $\boldsymbol{y}\to\boldsymbol{r}_i$ and $\|\boldsymbol{y}-\boldsymbol{r}_i\|_1<\epsilon$, we have $\mathrm{sign}(\boldsymbol{y}-\boldsymbol{r}_j)=\mathrm{sign}(\boldsymbol{r}_i-\boldsymbol{r}_j)$. Thus

$$
\begin{aligned}
s\cdot\mathrm{sign}(\boldsymbol{r}_i-\boldsymbol{r}_j)^\top((\boldsymbol{y}-\boldsymbol{r}_j)-(\boldsymbol{r}_i-\boldsymbol{r}_j)) \\
= s(\mathrm{sign}(\boldsymbol{y}-\boldsymbol{r}_j)^\top(\boldsymbol{y}-\boldsymbol{r}_j) - \mathrm{sign}(\boldsymbol{r}_i-\boldsymbol{r}_j)^\top(\boldsymbol{r}_i-\boldsymbol{r}_j)) \\
= s(\|\boldsymbol{y}-\boldsymbol{r}_j\|_1-\|\boldsymbol{r}_i-\boldsymbol{r}_j\|_1).
\end{aligned}
\tag{68}
$$

Let $\xi := \|\boldsymbol{y}-\boldsymbol{r}_j\|_1-\|\boldsymbol{r}_i-\boldsymbol{r}_j\|_1$ and define

$$
g(\xi) := \exp(s\xi) - s\xi - 1.
\tag{69}
$$

Then $g'(\xi)=s(\exp(s\xi)-1)$. Regardless of whether $s=1$ or $s=-1$, we have $g'(\xi)>0$ for $\xi>0$ and $g'(\xi)<0$ for $\xi<0$. Hence $g(\xi)\geqslant g(0)=0$. Substituting $\xi:=\|\boldsymbol{y}-\boldsymbol{r}_j\|_1-\|\boldsymbol{r}_i-\boldsymbol{r}_j\|_1$, we have

$$
\exp\left(s(\|\boldsymbol{y}-\boldsymbol{r}_j\|_1-\|\boldsymbol{r}_i-\boldsymbol{r}_j\|_1)\right)-1 \geqslant s(\|\boldsymbol{y}-\boldsymbol{r}_j\|_1-\|\boldsymbol{r}_i-\boldsymbol{r}_j\|_1)
\tag{70}
$$

for any $\boldsymbol{y}\to\boldsymbol{r}_i$. It verifies that $\partial\lambda_{ij}(\boldsymbol{r}_i)$ is a subgradient of $\lambda_{ij}$ at $\boldsymbol{r}_i$ by Definition 1.

**Case 4: $\mathcal{C}\neq\emptyset$ and $s=1$.** Directly computing the quotient in (21) gives

$$
\left[\exp(s(\sum_{d\notin\mathcal{C}}(|y_d-r_{j,d}|-|r_{i,d}-r_{j,d}|)+\sum_{d\in\mathcal{C}}|y_d-r_{j,d}|))-1-s(\sum_{d\notin\mathcal{C}}|y_d-r_{i,d}|)\right]/\sqrt{\sum_{d=1}^{\mathcal{D}}(y_d-r_{i,d})^2}.
\tag{71}
$$

Let $\boldsymbol{y}\to\boldsymbol{r}_i$, then $\mathrm{sign}(y_d-r_{j,d})=\mathrm{sign}(r_{i,d}-r_{j,d})$ for any $d\notin\mathcal{C}$. Combined with $r_{i,d}=r_{j,d}$ for any $d\in\mathcal{C}$, (71) becomes

$$
\begin{aligned}
&\left[\exp(s(\sum_{d\notin\mathcal{C}}(|y_d-r_{i,d}|)+\sum_{d\in\mathcal{C}}|y_d-r_{i,d}|))-1-s(\sum_{d\notin\mathcal{C}}|y_d-r_{i,d}|)\right]/\sqrt{\sum_{d=1}^{\mathcal{D}}(y_d-r_{i,d})^2} \\
&= \left[\exp(s(\sum_{d=1}^{\mathcal{D}}|y_d-r_{i,d}|))-1-s(\sum_{d\notin\mathcal{C}}|y_d-r_{i,d}|)\right]/\sqrt{\sum_{d=1}^{\mathcal{D}}(y_d-r_{i,d})^2}.
\end{aligned}
\tag{72}
$$

When $s=1$, we have

$$
\exp(\sum_{d=1}^{\mathcal{D}}|y_d-r_{i,d}|) \geqslant \sum_{d=1}^{\mathcal{D}}|y_d-r_{i,d}|+1 \geqslant \sum_{d\notin\mathcal{C}}|y_d-r_{i,d}|+1.
\tag{73}
$$

The first inequality follows from the same properties of $g(\xi)$ defined in (69) with $\xi:=\sum_{d=1}^{\mathcal{D}}|y_d-r_{i,d}|$. Therefore,

$$
\left[\exp(\sum_{d=1}^{\mathcal{D}}|y_d-r_{i,d}|)-1-\sum_{d\notin\mathcal{C}}|y_d-r_{i,d}|\right]/\sqrt{\sum_{d=1}^{\mathcal{D}}(y_d-r_{i,d})^2} \geqslant 0.
\tag{74}
$$

Hence $\partial\lambda_{ij}(\boldsymbol{r}_i)$ is a subgradient of $\lambda_{ij}$ at $\boldsymbol{r}_i$ by Definition 1. $\qquad\square$

### A.5 EXPERIMENTAL DETAILS

#### A.5.1 EXPERIMENTAL SETTING

The entire network is divided into two parts. Events within the last $10\%$ of the timeline form the second part, which is used as the prediction set. Meanwhile, $20\%$ of node pairs from the first part are randomly selected and then equally split into validation and test sets. The training set consists of the residual network that excludes any connections from the nodes in the validation and test sets. If a node pair does not have any connected periods during the training time but is selected for the prediction set, it is excluded from the entire network.

The next step is to generate labeled data for the connection prediction tasks. This involves dividing the timeline of each sampled node pair into segments based on their state values. For each segment, a random time $t$ selected to create a sample interval $[t-\varepsilon, t+\varepsilon]$, where $\varepsilon = 0.01T$ and $T$ represents the total length of the timeline. If a sample interval includes different states, it is considered as an indeterminate state and discarded.

The resulting samples are then categorized into "hard" and "simple" sets. The "hard" set contains samples from node pairs that have experienced at least one connected and one disconnected periods throughout the entire timeline (Huang et al., 2024b). In contrast, the "simple" set includes node pairs with only one state, making their state prediction relatively straightforward. Samples in the prediction set are categorized based on their connection status during the training time, following the setting in (Poursafaei et al., 2022).

For each training, validation, test, or prediction set, let $h$ be the size of the "hard" set, and define

$$k := \min\{1000, \text{connected set size}, \text{unconnected set size}\}. \tag{75}$$

Then each class (connected or disconnected) in the training, validation, test, or prediction set contains $h/2$ samples from the "hard" set, and $(k - h/2)$ samples from the remaining "hard" and "simple" sets. Performance is evaluated using AUC-ROC and AUC-PR scores, which assess true or false positive rates and precision-recall characteristics, respectively. All the experiments are repeated for 10 times to produce the mean and standard deviation results (mean±STD).

#### A.5.2 DATA SETS

The network is treated as undirected, with the finest available temporal granularity, such as measurements in seconds or milliseconds, used for the input timestamps. Additionally, the data sets are adapted to fit different competitors to ensure fair comparisons. For instance, dynamic networks are transformed into static weighted or unweighted networks by aggregating connections over the entire timeline for static competitors. Furthermore, since some competitors cannot handle non-connection events, these events are excluded from the evaluation.

We employ 11 synthetic and real-world benchmark data sets with diverse characteristics for our experiments, detailed below.

**Synthetic-$\alpha$.** This data set is generated using the sequential survival process outlined in Section 2, incorporating both connection and non-connection event times. The initial node positions and velocities are sampled using a procedure similar to that in (Çelikkanat et al., 2022). The global connectedness parameters are set as $\beta(-1) = 1$ and $\beta(1) = -1$. Velocities are sampled from a multivariate Gaussian distribution $\text{vec}(\boldsymbol{v}) \sim \mathcal{N}(\boldsymbol{0}, \theta^2\boldsymbol{\Sigma})$, where $\boldsymbol{\Sigma} := \text{diag}(\sigma_N\boldsymbol{1}_N, \sigma_B\boldsymbol{1}_B, \boldsymbol{\sigma}_{\mathcal{D}}\boldsymbol{1}_{\mathcal{D}}) \in \mathbb{R}^{N \cdot B \cdot \mathcal{D} \times N \cdot B \cdot \mathcal{D}}$. These hyperparameters are set as $\theta := 3$, $\sigma_N := 10^{-2}$, $\sigma_B := 10^{-6}$ and $\sigma_{\mathcal{D}} := 1$. Visualized key frames are shown in Figure A2, illustrating that the $\ell_1$ distance case leads to more between-group and longer-distance connections compared with the squared $\ell_2$ distance case.

**Synthetic-$\beta$.** The entire timeline is divided into 8 bins of equal length, with nodes randomly clustered into 10 groups. The probability of a node pair connection within the same group is set as $0.8$, while that of connections between different groups is set as $10^{-2}$. These connections persist until the next bin event occurs.

**Contacts** (Génois et al., 2015). This data set records interactions among individuals in an office building over a span of 9 days in 2013.

**HyperText** (Isella et al., 2011). It captures face-to-face interactions tracked by radio badges of participants during a 2.5-day conference.

**Infectious** (Isella et al., 2011). This data set includes another interaction network during an event in Dublin. In the Contacts, HyperText, and Infectious data sets, each timestamp represents a 20-second connection. Multiple connections within a two-minute window are treated as a single connection duration.

**Facebook** (Viswanath et al., 2009). This is a friendship network that indicates whether one user is on the friend list of another user. It is a large-scale network with about 82 thousand nodes and 200 million edges.

**NeurIPS** (Globerson et al., 2004). The original version tracks collaborations among authors for NeurIPS conference papers from 1989 to 2001. It is later extended to cover papers from 1988 to 2003 (Globerson et al., 2007). It is a large-scale network with about 5 thousand nodes and 1.2 million edges.

**US Legis.** (Poursafaei et al., 2022). It is a senate co-sponsorship graph that records social interactions between legislators from the US Senate. The edge weights indicate the number of times that two congress persons have co-sponsored a bill in a given congress.

**Can. Parl.** (Poursafaei et al., 2022). It is a dynamic political network that records the interactions between Canadian Members of Parliaments (MPs) from 2006 to 2019. Each node represents one MP for an electoral district, and each edge is formed if both of the corresponding two MPs voted *yes* on a bill. The edge weights indicate the number of times that one MP voted *yes* for another MP in a year.

**Wikipedia** (Poursafaei et al., 2022). It is a co-editing network on Wikipedia pages over one month. Editors and wiki pages are nodes, while one edge represents that a given user edits a page at a specific timestamp. The task is to predict with which wiki page a user will interact at a given time.

**Reddit** (Poursafaei et al., 2022). It is an interaction network for users and subreddits from 2008 to 2010. Both users and subreddits are nodes, while each edge indicates that a user posted on a subreddit at a given time. The task is to learn the interaction frequency towards the subreddits of a user over the next week.

### A.5.3    COMPARED METHODS

To assess the performance of the proposed $\ell_1$LD-CTGR method, we compare it against 8 leading methods with static, dynamic, and continuous-time graph representations. These methods serve as strong baselines for evaluating the effectiveness of our approach. We adopt all the default settings of these methods in the experiments.

**Node2Vec** (Grover & Leskovec, 2016). Node2Vec introduces a flexible concept for defining the neighborhood of a node and designs a biased random walk algorithm to efficiently explore various neighborhoods. Its default embedding dimensionality is $\mathcal{D} = 128$.

**CTDNE** (Nguyen et al., 2018). Continuous-Time Dynamic Network Embedding (CTDNE) extends random walk methods to dynamic networks, which capture and integrate temporal information into network embeddings. Its default embedding dimensionality is $\mathcal{D} = 128$.

**HTNE** (Zuo et al., 2018). Hawkes Temporal Network Embedding (HTNE) adopts a Hawkes process based approach to capture the influence of historical neighbors on the current neighbors into temporal network embeddings. Its default embedding dimensionality is $\mathcal{D} = 128$.

**PIVEM** (Çelikkanat et al., 2022). Piece-wise Interpolation for Temporal Network Embedding (PIVEM) uses piece-wise linear interpolations to approximate temporal node evolution. Its project link[2] indicates that the default embedding dimensionality is $\mathcal{D} = 10$. It uses $B = 100$ bins in the timeline segmentation.

**TCL** (Wang et al., 2021; Yu et al., 2023). It applies contrastive learning to representation learning on dynamic graphs, which builds on a graph-topology-aware transformer. Its default embedding dimensionality is $\mathcal{D} = 322$.

---

[2]https://abdcelikkanat.github.io/projects/pivem/

**GraphMixer** (Cong et al., 2023; Yu et al., 2023). It consists of three components: a link-decoder that treats temporal links, a node-encoder that summarizes node information, and a link classifier that performs link prediction. Its default embedding dimensionality is $\mathcal{D} = 322$.

**DyGFormer** (Yu et al., 2023). It is a Transformer-based architecture that only needs to learn from the historical first-hop interactions of nodes by a neighbor co-occurrence encoding scheme and a patching technique.

**GRASSP** (Çelikkanat et al., 2024). It employs the squared $\ell_2$ latent distance based sequential survival process to model continuous-time graph evolution. It uses the ultra-low-dimensional embedding with $\mathcal{D} = 2$ and the same number of bins $B = 100$ as that of PIVEM.

**$\ell_1$LD-CTGR (ours)**. It also uses the ultra-low-dimensional embedding with $\mathcal{D} = 2$, and the same number of bins $B = 100$ as that of PIVEM or GRASSP, to be consistent.

### A.5.4 RUNTIME COMPARISON BETWEEN GRASSP AND $\ell_1$LD-CTGR (TABLE A1)

Table A1: Average runtime (in seconds) per epoch of GRASSP, $\ell_1$LD-CTGR, and the $\ell_2$ distance version ($H = 1000$) with ultra-low-dimensionality $\mathcal{D} = 2$ on different data sets (mean $\pm$ STD). The training criterion of GRASSP is followed, which is a three-stage 300-epoch procedure. Results are conducted on a device with an Intel(R) Xeon(R) Gold 6330 CPU, 1 TB RAM, and eight NVIDIA A100 GPUs. Both GRASSP and $\ell_1$LD-CTGR have the same order of actual runtime.

| Data Set | GRASSP | $\ell_1$LD-CTGR | $\ell_2$ Distance |
|---|---|---|---|
| Synthetic-$\alpha$ | $1.93E-4 \pm 8.97E-6$ | $1.94E-4 \pm 9.44E-6$ | $2.29E-2 \pm 1.13E-3$ |
| Synthetic-$\beta$ | $1.64E-4 \pm 7.15E-6$ | $1.66E-4 \pm 7.81E-6$ | $1.94E-2 \pm 9.34E-4$ |
| Contacts | $3.21E-3 \pm 1.60E-4$ | $3.09E-3 \pm 1.45E-4$ | $3.80E-1 \pm 1.70E-2$ |
| HyperText | $6.18E-3 \pm 2.98E-4$ | $6.22E-3 \pm 2.90E-4$ | $7.10E-1 \pm 3.53E-2$ |
| Infectious | $6.15E-3 \pm 3.00E-4$ | $6.15E-3 \pm 2.79E-4$ | $7.26E-1 \pm 3.36E-2$ |
| Facebook | $4.49 \pm 2.18E-1$ | $4.55 \pm 2.33E-1$ | $531 \pm 28.4$ |
| NeurIPS | $2.74E-1 \pm 1.41E-2$ | $2.77E-1 \pm 1.37E-2$ | $33.8 \pm 1.56$ |
| US Legis. | $1.86E-2 \pm 9.39E-4$ | $1.76E-2 \pm 9.10E-4$ | $2.14 \pm 1.08E-1$ |
| Can. Parl. | $2.49E-2 \pm 1.20E-3$ | $2.44E-2 \pm 1.31E-3$ | $2.93 \pm 1.50E-1$ |

### A.5.5 ROBUSTNESS OF $\ell_1$LD-CTGR TO BIN NUMBER AND EMBEDDING DIMENSIONALITY

We examine the robustness of $\ell_1$LD-CTGR to bin number (default $B = 100$) and embedding dimensionality (default $\mathcal{D} = 2$). We first fix $\mathcal{D} = 2$ and let $B$ change in $[92, 108]$, and then fix $B = 100$ and let $\mathcal{D}$ change in $[2, 6]$. The corresponding results on the Contacts data set are shown in Tables A3 and A4, respectively. They indicate that $\ell_1$LD-CTGR is robust to small changes of these hyperparameters. Moreover, we directly compare the performance of $\ell_1$LD-CTGR, GRASSP, and $\ell_2$ Distance on the Contacts and Infectious data sets in Table A4. Results show that $\ell_1$LD-CTGR achieves the best performance in most cases, while $\ell_2$ Distance generally underperforms $\ell_1$LD-CTGR and GRASSP to a certain extent.

### A.5.6 NETWORK RECONSTRUCTION PERFORMANCE (TABLE A5)

### A.5.7 CONTINUOUS-TIME VISUALIZATION (FIGURE A2)

Following (Çelikkanat et al., 2024), we visualize the evolutions of the $\ell_1$ latent distance and the squared $\ell_2$ distance based Synthetic-$\alpha$ networks in Figure A2.

Table A2: Average runtime (in seconds) per epoch of GRASSP and $\ell_1$LD-CTGR with higher dimensionalities on different data sets (mean ± STD).

| Data Set | GRASSP | $\ell_1$LD-CTGR |
|---|---|---|
| | $\mathcal{D} = 10$ | |
| Synthetic-$\alpha$ | $1.98E - 4 \pm 9.20E - 6$ | $1.99E - 4 \pm 9.67E - 6$ |
| Synthetic-$\beta$ | $1.68E - 4 \pm 7.33E - 6$ | $1.69E - 4 \pm 8.00E - 6$ |
| Contacts | $3.29E - 3 \pm 1.64E - 4$ | $3.16E - 3 \pm 1.50E - 4$ |
| HyperText | $6.33E - 3 \pm 3.05E - 4$ | $6.36E - 3 \pm 2.99E - 4$ |
| Infectious | $6.32E - 3 \pm 3.07E - 4$ | $6.31E - 3 \pm 2.86E - 4$ |
| Facebook | $4.61 \pm 2.23E - 1$ | $4.66 \pm 2.39E - 1$ |
| NeurIPS | $2.81E - 1 \pm 1.45E - 2$ | $2.85E - 1 \pm 1.40E - 2$ |
| US Legis. | $1.90E - 2 \pm 9.69E - 4$ | $1.80E - 2 \pm 9.34E - 4$ |
| Can. Parl. | $2.55E - 2 \pm 1.23E - 3$ | $2.49E - 2 \pm 1.34E - 3$ |
| | $\mathcal{D} = 100$ | |
| Synthetic-$\alpha$ | $2.21E - 4 \pm 1.02E - 5$ | $2.43E - 4 \pm 1.20E - 5$ |
| Synthetic-$\beta$ | $1.88E - 4 \pm 8.20E - 6$ | $2.08E - 4 \pm 1.01E - 5$ |
| Contacts | $3.72E - 3 \pm 1.82E - 4$ | $3.92E - 3 \pm 1.86E - 4$ |
| HyperText | $7.12E - 3 \pm 3.44E - 4$ | $7.93E - 3 \pm 3.73E - 4$ |
| Infectious | $7.02E - 3 \pm 3.44E - 4$ | $7.83E - 3 \pm 3.60E - 4$ |
| Facebook | $5.09 \pm 2.47E - 1$ | $5.88 \pm 3.05E - 1$ |
| NeurIPS | $3.14E - 1 \pm 1.62E - 2$ | $3.54E - 1 \pm 1.71E - 2$ |
| US Legis. | $2.18E - 2 \pm 1.07E - 5$ | $2.23E - 2 \pm 1.19E - 3$ |
| Can. Parl. | $2.85E - 2 \pm 1.37E - 3$ | $3.09E - 2 \pm 1.70E - 3$ |

Table A3: Performance of $\ell_1$LD-CTGR with respect to different bin numbers on Contact (mean of 10 repetitions).

| Bin. Num. | Completion | | Reconstruction | | Prediction | |
|---|---|---|---|---|---|---|
| | ROC | PR | ROC | PR | ROC | PR |
| 92 | 0.660 | 0.714 | 0.636 | 0.646 | 0.772 | 0.726 |
| 94 | 0.655 | 0.709 | 0.641 | 0.651 | 0.767 | 0.721 |
| 96 | 0.645 | 0.694 | 0.661 | 0.656 | 0.757 | 0.711 |
| 98 | 0.650 | 0.699 | 0.676 | 0.666 | 0.762 | 0.716 |
| 100 | 0.680 | 0.724 | 0.676 | 0.676 | 0.767 | 0.721 |
| 102 | 0.635 | 0.689 | 0.636 | 0.641 | 0.752 | 0.706 |
| 104 | 0.640 | 0.694 | 0.631 | 0.636 | 0.757 | 0.711 |
| 106 | 0.650 | 0.704 | 0.636 | 0.641 | 0.762 | 0.716 |
| 108 | 0.645 | 0.699 | 0.636 | 0.641 | 0.757 | 0.716 |

Table A4: Performance of $\ell_1$LD-CTGR, GRASSP, and the $\ell_2$ distance version ($H = 1000$) with respect to different dimensionalities on Contact and Infectious (mean of 10 repetitions).

| Dim. Size | | Contact | | | | | | Infectious | | | | | |
|---|---|---|---|---|---|---|---|---|---|---|---|---|---|
| | | Completion | | Reconstruction | | Prediction | | Completion | | Reconstruction | | Prediction | |
| | | ROC | PR | ROC | PR | ROC | PR | ROC | PR | ROC | PR | ROC | PR |
| | | | | | | | $\ell_1$LD-CTGR | | | | | | |
| 2 | Mean | 0.680 | 0.724 | 0.676 | 0.676 | 0.767 | 0.721 | 0.756 | 0.779 | 0.861 | 0.832 | 0.901 | 0.888 |
| | STD | 0.017 | 0.028 | 0.016 | 0.021 | 0.018 | 0.018 | 0.017 | 0.017 | 0.021 | 0.019 | 0.016 | 0.016 |
| 3 | Mean | 0.664 | 0.704 | 0.657 | 0.680 | 0.737 | 0.699 | 0.753 | 0.765 | 0.846 | 0.817 | 0.885 | 0.883 |
| | STD | 0.016 | 0.009 | 0.024 | 0.020 | 0.016 | 0.023 | 0.013 | 0.018 | 0.018 | 0.020 | 0.020 | 0.016 |
| 4 | Mean | 0.655 | 0.686 | 0.638 | 0.668 | 0.733 | 0.682 | 0.745 | 0.767 | 0.858 | 0.820 | 0.888 | 0.875 |
| | STD | 0.019 | 0.018 | 0.021 | 0.022 | 0.023 | 0.016 | 0.019 | 0.015 | 0.015 | 0.019 | 0.024 | 0.016 |
| 5 | Mean | 0.647 | 0.696 | 0.636 | 0.654 | 0.744 | 0.704 | 0.736 | 0.748 | 0.859 | 0.811 | 0.882 | 0.876 |
| | STD | 0.020 | 0.018 | 0.023 | 0.015 | 0.019 | 0.021 | 0.022 | 0.024 | 0.019 | 0.014 | 0.016 | 0.015 |
| 6 | Mean | 0.633 | 0.690 | 0.674 | 0.671 | 0.777 | 0.735 | 0.735 | 0.737 | 0.869 | 0.830 | 0.879 | 0.886 |
| | STD | 0.028 | 0.016 | 0.019 | 0.018 | 0.012 | 0.017 | 0.013 | 0.022 | 0.016 | 0.019 | 0.017 | 0.020 |
| | | | | | | | GRASSP | | | | | | |
| 2 | Mean | 0.670 | 0.714 | 0.589 | 0.634 | 0.763 | 0.714 | 0.728 | 0.711 | 0.738 | 0.708 | 0.898 | 0.861 |
| | STD | 0.016 | 0.025 | 0.013 | 0.019 | 0.016 | 0.020 | 0.029 | 0.028 | 0.018 | 0.016 | 0.015 | 0.017 |
| 3 | Mean | 0.659 | 0.704 | 0.605 | 0.632 | 0.734 | 0.703 | 0.718 | 0.703 | 0.729 | 0.698 | 0.873 | 0.885 |
| | STD | 0.022 | 0.025 | 0.023 | 0.018 | 0.021 | 0.015 | 0.024 | 0.013 | 0.014 | 0.015 | 0.004 | 0.027 |
| 4 | Mean | 0.645 | 0.689 | 0.593 | 0.639 | 0.739 | 0.679 | 0.713 | 0.706 | 0.734 | 0.713 | 0.885 | 0.877 |
| | STD | 0.021 | 0.015 | 0.027 | 0.010 | 0.019 | 0.018 | 0.021 | 0.021 | 0.015 | 0.019 | 0.019 | 0.023 |
| 5 | Mean | 0.656 | 0.689 | 0.568 | 0.612 | 0.736 | 0.689 | 0.702 | 0.686 | 0.732 | 0.693 | 0.876 | 0.851 |
| | STD | 0.025 | 0.025 | 0.015 | 0.020 | 0.013 | 0.014 | 0.019 | 0.027 | 0.015 | 0.021 | 0.018 | 0.025 |
| 6 | Mean | 0.652 | 0.685 | 0.562 | 0.626 | 0.744 | 0.705 | 0.719 | 0.702 | 0.729 | 0.700 | 0.887 | 0.862 |
| | STD | 0.014 | 0.020 | 0.022 | 0.025 | 0.027 | 0.020 | 0.012 | 0.016 | 0.014 | 0.029 | 0.016 | 0.023 |
| | | | | | | | $\ell_2$ Distance | | | | | | |
| 2 | Mean | 0.605 | 0.625 | 0.614 | 0.606 | 0.692 | 0.623 | 0.688 | 0.691 | 0.721 | 0.694 | 0.834 | 0.811 |
| | STD | 0.038 | 0.034 | 0.035 | 0.029 | 0.022 | 0.024 | 0.021 | 0.013 | 0.017 | 0.021 | 0.008 | 0.012 |
| 3 | Mean | 0.594 | 0.639 | 0.552 | 0.617 | 0.644 | 0.593 | 0.678 | 0.677 | 0.716 | 0.688 | 0.809 | 0.803 |
| | STD | 0.028 | 0.027 | 0.027 | 0.023 | 0.023 | 0.027 | 0.023 | 0.019 | 0.023 | 0.018 | 0.019 | 0.013 |
| 4 | Mean | 0.599 | 0.589 | 0.559 | 0.574 | 0.655 | 0.607 | 0.676 | 0.673 | 0.719 | 0.701 | 0.814 | 0.804 |
| | STD | 0.026 | 0.023 | 0.034 | 0.026 | 0.020 | 0.026 | 0.019 | 0.021 | 0.024 | 0.017 | 0.017 | 0.023 |
| 5 | Mean | 0.567 | 0.626 | 0.544 | 0.569 | 0.624 | 0.623 | 0.677 | 0.675 | 0.719 | 0.698 | 0.807 | 0.803 |
| | STD | 0.025 | 0.026 | 0.022 | 0.020 | 0.026 | 0.023 | 0.024 | 0.014 | 0.021 | 0.014 | 0.019 | 0.020 |
| 6 | Mean | 0.530 | 0.609 | 0.632 | 0.587 | 0.678 | 0.647 | 0.673 | 0.666 | 0.720 | 0.709 | 0.818 | 0.810 |
| | STD | 0.030 | 0.022 | 0.029 | 0.023 | 0.022 | 0.019 | 0.013 | 0.018 | 0.027 | 0.029 | 0.014 | 0.017 |

Table A5: Performance of different methods for network reconstruction (in-sample) across diverse data sets (mean±STD).

| Data Set | | Node2Vec | CTDNE | HTNE | PIVEM | TCL | GraphMixer | DyGFormer | GRASSP | $\ell_1$LD-CTGR |
|---|---|---|---|---|---|---|---|---|---|---|
| Synthetic-$\alpha$ | ROC | 0.524 | 0.550 | 0.535 | 0.567 | 0.281 | 0.300 | 0.436 | 0.577 | **0.793** |
| | | ±0.013 | ±0.016 | ±0.021 | ±0.018 | ±0.042 | ±0.038 | ±0.020 | ±0.028 | **±0.038** |
| | PR | 0.538 | 0.555 | 0.602 | 0.642 | 0.400 | 0.412 | 0.446 | 0.509 | **0.708** |
| | | ±0.023 | ±0.027 | ±0.019 | ±0.016 | ±0.009 | ±0.027 | ±0.022 | ±0.029 | **±0.039** |
| Synthetic-$\beta$ | ROC | 0.541 | 0.493 | 0.535 | 0.531 | 0.528 | 0.448 | 0.536 | **0.843** | 0.610 |
| | | ±0.006 | ±0.008 | ±0.008 | ±0.006 | ±0.061 | ±0.006 | ±0.018 | **±0.015** | ±0.021 |
| | PR | 0.545 | 0.557 | 0.591 | 0.536 | 0.621 | 0.573 | 0.616 | **0.756** | 0.554 |
| | | ±0.004 | ±0.006 | ±0.006 | ±0.006 | ±0.054 | ±0.016 | ±0.020 | **±0.011** | ±0.017 |
| Contacts | ROC | 0.674 | 0.508 | 0.555 | 0.539 | 0.594 | 0.539 | 0.596 | 0.589 | **0.676** |
| | | ±0.011 | ±0.021 | ±0.011 | ±0.012 | ±0.007 | ±0.012 | ±0.016 | ±0.013 | **±0.016** |
| | PR | 0.657 | 0.570 | 0.563 | 0.567 | 0.616 | 0.614 | 0.642 | 0.634 | **0.676** |
| | | ±0.016 | ±0.019 | ±0.013 | ±0.009 | ±0.016 | ±0.013 | ±0.019 | ±0.019 | **±0.021** |
| HyperText | ROC | 0.589 | 0.486 | 0.619 | 0.560 | 0.602 | 0.662 | 0.681 | 0.607 | **0.694** |
| | | ±0.006 | ±0.014 | ±0.011 | ±0.004 | ±0.010 | ±0.006 | ±0.014 | ±0.006 | **±0.011** |
| | PR | 0.569 | 0.542 | 0.624 | 0.572 | 0.627 | 0.663 | 0.640 | 0.580 | **0.691** |
| | | ±0.008 | ±0.013 | ±0.007 | ±0.004 | ±0.007 | ±0.003 | ±0.015 | ±0.009 | **±0.014** |
| Infectious | ROC | 0.781 | 0.501 | 0.851 | 0.613 | 0.811 | 0.804 | 0.839 | 0.738 | **0.861** |
| | | ±0.003 | ±0.009 | ±0.011 | ±0.005 | ±0.001 | ±0.001 | ±0.012 | ±0.018 | **±0.021** |
| | PR | 0.742 | 0.566 | 0.819 | 0.630 | 0.812 | 0.801 | **0.848** | 0.708 | 0.832 |
| | | ±0.008 | ±0.011 | ±0.009 | ±0.007 | ±0.004 | ±0.001 | **±0.013** | ±0.016 | ±0.019 |
| Facebook | ROC | 0.506 | 0.473 | 0.445 | 0.482 | 0.510 | 0.500 | 0.502 | 0.500 | **0.612** |
| | | ±0.002 | ±0.005 | ±0.003 | ±0.002 | ±0.002 | ±0.009 | ±0.010 | ±0.000 | **±0.004** |
| | PR | 0.515 | 0.489 | 0.481 | **0.625** | 0.520 | 0.530 | 0.540 | 0.500 | 0.588 |
| | | ±0.004 | ±0.005 | ±0.003 | **±0.003** | ±0.001 | ±0.006 | ±0.011 | ±0.000 | ±0.004 |
| NeurIPS | ROC | 0.433 | 0.489 | 0.431 | 0.510 | 0.635 | 0.634 | **0.688** | 0.548 | 0.528 |
| | | ±0.004 | ±0.011 | ±0.011 | ±0.009 | ±0.001 | ±0.001 | **±0.020** | ±0.018 | ±0.009 |
| | PR | 0.476 | 0.541 | 0.448 | 0.525 | 0.580 | 0.578 | **0.613** | 0.506 | 0.501 |
| | | ±0.004 | ±0.015 | ±0.008 | ±0.008 | ±0.004 | ±0.006 | **±0.018** | ±0.025 | ±0.008 |
| US Legis. | ROC | 0.493 | 0.478 | 0.490 | 0.525 | 0.750 | 0.766 | **0.866** | 0.662 | 0.767 |
| | | ±0.003 | ±0.011 | ±0.017 | ±0.012 | ±0.005 | ±0.014 | **±0.016** | ±0.012 | ±0.014 |
| | PR | 0.510 | 0.524 | 0.576 | 0.561 | 0.515 | 0.700 | **0.856** | 0.588 | 0.712 |
| | | ±0.004 | ±0.013 | ±0.020 | ±0.012 | ±0.004 | ±0.008 | **±0.017** | ±0.018 | ±0.012 |
| Can. Parl. | ROC | 0.701 | 0.479 | 0.583 | 0.508 | 0.687 | 0.761 | **0.906** | 0.593 | 0.826 |
| | | ±0.004 | ±0.009 | ±0.011 | ±0.009 | ±0.004 | ±0.009 | **±0.010** | ±0.013 | ±0.004 |
| | PR | 0.649 | 0.542 | 0.643 | 0.527 | 0.634 | 0.743 | **0.911** | 0.614 | 0.793 |
| | | ±0.004 | ±0.009 | ±0.015 | ±0.014 | ±0.005 | ±0.005 | **±0.011** | ±0.008 | ±0.003 |
| Wikipedia | ROC | 0.312 | 0.480 | 0.430 | 0.620 | 0.852 | 0.868 | 0.760 | 0.705 | **0.870** |
| | | ±0.010 | ±0.011 | ±0.012 | ±0.009 | ±0.006 | ±0.007 | ±0.011 | ±0.008 | **±0.005** |
| | PR | 0.321 | 0.453 | 0.471 | 0.680 | 0.890 | **0.908** | 0.805 | 0.668 | 0.838 |
| | | ±0.009 | ±0.012 | ±0.013 | ±0.010 | ±0.007 | **±0.008** | ±0.012 | ±0.009 | ±0.006 |
| Reddit | ROC | 0.467 | 0.488 | 0.501 | 0.653 | 0.750 | 0.766 | 0.790 | 0.707 | **0.801** |
| | | ±0.012 | ±0.013 | ±0.014 | ±0.016 | ±0.010 | ±0.010 | ±0.012 | ±0.011 | **±0.009** |
| | PR | 0.456 | 0.465 | 0.530 | 0.681 | 0.769 | 0.767 | 0.801 | 0.689 | **0.803** |
| | | ±0.011 | ±0.012 | ±0.013 | ±0.014 | ±0.010 | ±0.010 | ±0.012 | ±0.011 | **±0.009** |

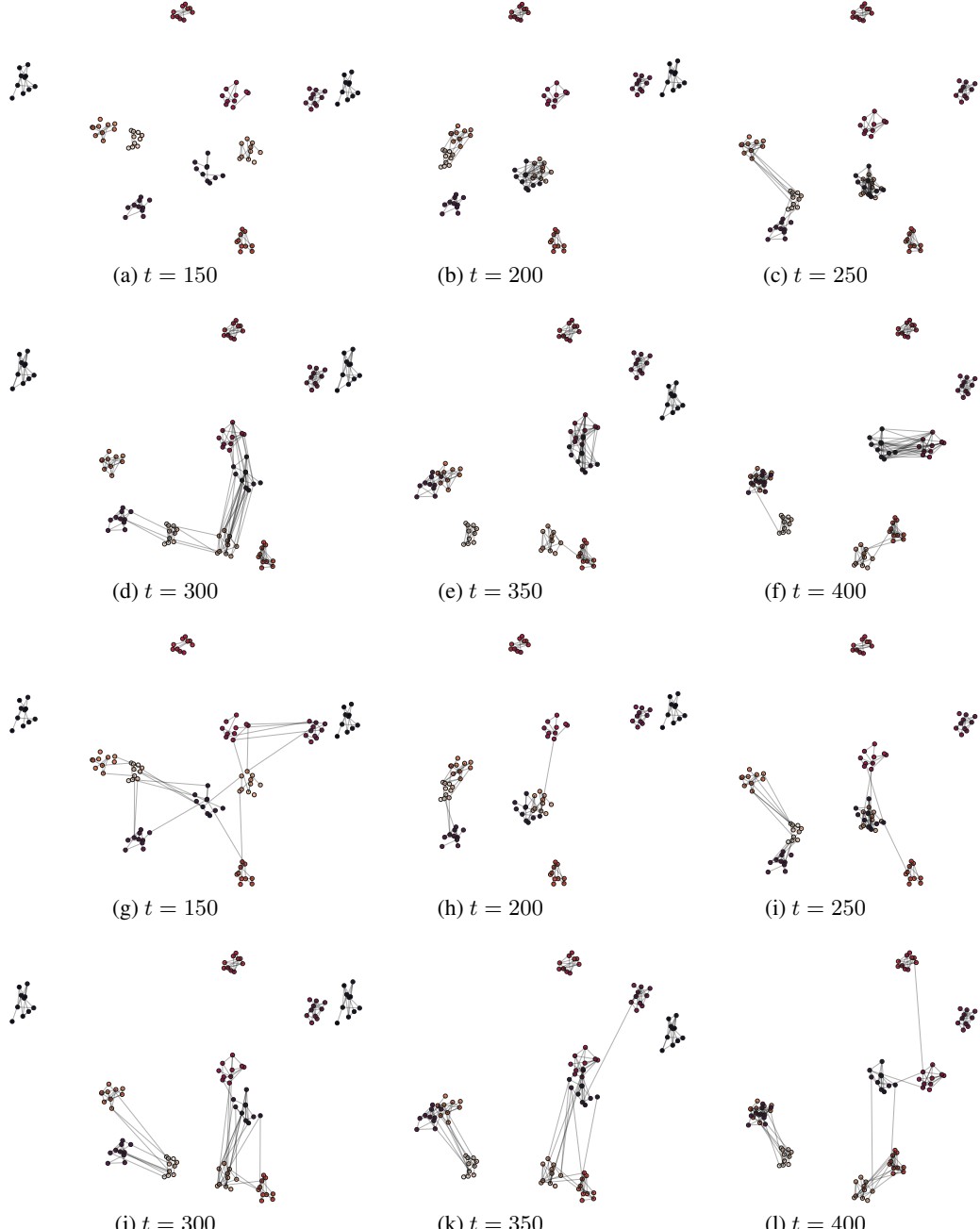

(a) $t = 150$

(b) $t = 200$

(c) $t = 250$

(d) $t = 300$

(e) $t = 350$

(f) $t = 400$

(g) $t = 150$

(h) $t = 200$

(i) $t = 250$

(j) $t = 300$

(k) $t = 350$

(l) $t = 400$

Figure A2: Continuous-time graph representation via the latent space on Synthetic-$\alpha$. (a)$\sim$(f): the squared $\ell_2$ distance case. (g)$\sim$(l): the $\ell_1$ distance case.

