# OpenReview forum: "$\ell_1$ Latent Distance based Continuous-time Graph Representation"
_ICLR.cc/2026/Conference — ICLR 2026 Poster_

### Official Review · Reviewer_51D8 · 2025-10-26

**Soundness:** 4
**Presentation:** 2
**Contribution:** 4
**Rating:** 8
**Confidence:** 3

**Summary:**

This paper proposes a latent position model for continuous-time using a hazard function based on $\ell_1$ distance between latent positions motivated by issues regarding the triangle inequality from previous methods using squared $\ell_2$ distance. The model is benchmarked against continuous-time graph representation models and other common graph representation techniques against commonly used synthetic and real dynamic networks with impressive results in incredibly low embedding dimension.

**Strengths:**

The technical work in this paper is strongly motivated and the argument for using $\ell_1$ distance for social networks, highlighting issues with using the squared $\ell^2$ distance used in GRASSP. While the computations are different in nature to GRASSP, the new approach has the same complexity cost as the previous approach, usurping the technique for incredibly low dimensional embeddings ($D = 2$).

**Weaknesses:**

The experimental results lack explanation. For instance, it is not explained how comparison continuous-time methods based on instantaneous edge models (CTDNE/HTNE) are applied to the persistent edge datasets. These methods are trying to model a different problem, so it not surprising they underperform. However, $\ell_1$LD-CTGR often outperforms GRASSP which is a like-minded comparison.

More information than what is given in Appendix A.5.1 about in-sample, out-of-sample and across-sample evaluations is needed. I understand the intended aims of the three approaches, but I don't see how it relates to the language of 'simple' and 'hard' sets. More details are needed to make this a repeatable experiment.

**Questions:**

Some datasets have worse performance when you move from in-sample to out-of-sample to across-sample assessment, for example, synthetic-$\alpha$. In general, I would expect the in-sample to be the easiest assessment as it is looking within training data, but it occasionally underperforms compared to the other assessments. Can the authors provide some intuition as to why this phenomenon occurs?

Does using the $\ell_1$LD-CTGR technique with dimension $D > 2$ lead to any improvement in performance? If not, this would further motivate the argument for using the computational efficient $D = 2$ embedding.

---

> ### Author Response · Authors · 2025-11-18
>
> We sincerely thank the reviewer for the in-depth evaluation and valuable feedback on this paper. We are pleased that the reviewer generally recognizes the core motivation of our work (addressing the theoretical problem of the squared L2 distance violating the triangle inequality) and the theoretical completeness (deriving the closed-form solution for the L1 integral and handling non-differentiability).
>
> **1.** The in-sample (reconstruction) task  requires the model to reproduce the _entire_ interaction history of a node pair. If this history is very complex (e.g., frequent state changes), its difficulty may be higher than generalizing to new node pairs (completion).
>
> **2.** We have already tested the performance for $D=2$ to $D=6$ in Table A4 (appendix). The results show that performance is robust to changes in $D$, and $D=2$ or $D=3$ often achieves the best or near-best performance. This further supports that our focus on the ultra-low dimensionality ($D=2$) is reasonable.
>
> **Replies to Other Concerns in Weaknesses.** For baselines based on instantaneous events like CTDNE and HTNE, we follow standard practice (the same as how GRASSP does) by treating the start and end times of persistent edges as events so that they can run. Although they model a different problem, this explains their poor performance and highlights the necessity of models like GRASSP and the proposed L1LD-CTGR, which can handle persistence.
>
> About "in-sample/out-of-sample/across-sample" assessment and "simple/hard" sets: This is actually terminology from ref. (Celikkanat et al., 2024). First, all samples are categorized as simple or hard samples. Second, the training, validation, test, and prediction sets are established by the same rule of picking simple/hard samples, to ensure that they have an appropriate proportion of simple/hard samples. Third, the in-sample, out-of-sample, and across-sample assessments are implemented by using the above four sets.

---

> > ### Comment · Reviewer_51D8 · 2025-11-20
> >
> > Thank you for your response. I am content to maintain my high rating of this paper.

---

> > > ### Author Response · Authors · 2025-11-25
> > >
> > > Dear reviewer,
> > >
> > > Thank you for the reply. We are glad that we have addressed your concerns.
> > >
> > > Authors

---

### Official Review · Reviewer_Y6yo · 2025-10-29

**Soundness:** 3
**Presentation:** 3
**Contribution:** 3
**Rating:** 4
**Confidence:** 5

**Summary:**

This work expands upon the recent GRASSP framework for the modeling of continuous time dynamics using the sequential survival process and piecewise velocity modeling procedure developed in (Celikkanat et al., 2024), Specifically, the paper here makes the following contribution:

1)	Advance the modeling to a true metric embedding model based on the l1 distance as opposed to the squared Euclidean distance used due to computational convenience in (Celikkanat et al., 2024).

2)	Demonstrate that this procedure despite added complexity of inference in practice have same scaling properties with a subgradient procedure proposed well optimizing the L1 distance while preserving other essential properties of GRASSP.

3)	Extensively evaluate the procedure and contrast the framework to more approaches than previously considered finding that the L1 distance based framework has enhanced performance when compared to the previous squared L2 distance embedding framework of GRASSP – pointing towards metric properties enhancing performance.

**Strengths:**

•	The paper is very well written and clear in its presentation.

•	The contribution is sound and valid and includes a detailed experimental comparison to existing methodologies.

•	The inference procedure is non-trivial and ensures metric properties in latent distance based continuous time link persistent networks.

Originality: The paper expands upon the GRASSP framework to make the embeddings truly metric by use of the L1 norm as opposed to relying on the squared Euclidean distance. The experimentation closely follows the GRASSP framework but is extensive. The contribution is sufficiently original.

Quality: The developed methodology is non-trivial and the paper well written and the experimentation extensive. It would however improve the paper to compare the L2-norm metric version with the proposed L1-norm metric version. It is unclear what comes from change of metric and what is the result of non-metric squaring the L2-norm. The metric L2-norm would be less computational attractive but could be implemented by use of numeric integration instead of relying on analytical expressions of the integral representations.

Clarity: The paper is clear and well presented.

Significance: The approach appears useful and superior to using the squared L2 distance with results well supporting the utility of the proposed approach.

**Weaknesses:**

•	The novelty of the paper is somewhat limited but valid expanding the prior work on GRASSP to encompass true metric embeddings as opposed to relying on computational convenient squared L2 distances.

•	It is unclear how difficult in practice using the standard Euclidean distance would be. I.e., the integral rather than being analytically solved could here be solved by numeric integration which would likely not be very heavy in computation and potentially work well in practice. This would strengthen the paper to consider.

**Questions:**

Experimentally the superiority of the L1 distance metric over squared Euclidean distance is shown. The method compared against have been run in their default setting. It is unclear if the proposed method have been more extensively tuned in hyperparameters in terms of learning rates etc. Only used dimensionality is discussed. Please clarify how learning rate etc. were tuned for the methods.

Could the L2-metric approach using a truly Euclidean metric as opposed to squared Euclidean distance representation be feasibly implemented using numeric integration as opposed to analytical expressions using the squared distance? – and how would such numeric integration based metric approach compare to the presented L1 approach?
This would better position the paper as it is unclear in the current experimentation what the influence of having a metric vs. non-metric embedding implies (this would be comparing squared L2 to conventional L2-norm (metric) - which could have been addressed by accurate numeric integration potentially on smaller toy problems) and what is the effect of L2-norm (metric) vs. L1-norm (metric) presently invoked. These different norms will by themselves substantially influence the results and here the L1-norm may also have benefits over the L2-norm which cannot currently be assessed. This would strengthen the paper to address and help further position the contribution and understanding the merits of the L1-norm.

Given the above I consider this a borderline paper but would be willing to increase my score.

---

> ### Author Response · Authors · 2025-11-18
>
> We sincerely thank the reviewer for the in-depth evaluation and valuable feedback on this paper. We are pleased that the reviewer generally recognizes the theoretical completeness (deriving the closed-form solution for the L1 integral and handling non-differentiability) and the comprehensiveness of the experiments.
>
> **1.** We use the Adam optimizer [a] and set the learning rate to $0.1$ in model learning. We do not tune the learning rate because Adam is itself an adaptive learning mechanism, known for its robustness and often performs well across various tasks.
>
> **2.** We add a detailed analysis on the numerical integral of the L2 distance in Section 2.2, around Eq. 10 and Eq. 11 of the revised manuscript. In brief, it has two main drawbacks: 1) It causes a significant increase of computational complexity from $\tilde{O}(D)$ of both L1LD-CTGR and GRASSP to $O(HD)$, where $H$ denotes the number of subintervals into which the entire integration interval is divided, and $D$ is the embedding dimensionality. 2) It causes an approximation error compared to the exact integral, which can be amplified by composite function relationships. To achieve an acceptable approximating accuracy, $H$ is generally set to $1000$ or even larger, then the computational complexity is significantly increased. Worse still, such an approximating accuracy cannot bring about good performance. We add runtime and representation learning experiments for the L2 distance version ($H=1000$) in Tables A1 and A4 of the revised manuscript, respectively. Table A1 shows that the L2 distance version requires much more runtime than L1LD-CTGR and GRASSP, while Table A4 shows that the L2 distance version performs much worse than L1LD-CTGR in representation learning. Hence the L2 distance version is inefficient and ineffective in our task.
>
> [a] Kingma, D. P. and Ba, J. Adam: A method for stochastic optimization. In International Conference on Learning Representations, 2015.

---

> > ### Comment · Reviewer_Y6yo · 2025-11-21
> >
> > I thank the authors for including the requested experimentation including the L2-metric distance by numeric integration. I agree that the runtime will be much worse and that this of course is a drawback. I am however surprised that H=1000 fails to produce reasonable results in network completion, reconstruction, and prediction. This is however also only shown for one dataset (contacts) and not including the L2-squared analytic results for comparison.
> >
> > I would appreciate to see the results for more networks (if necessary considering only the networks with similar compute requirements as contacts (as indicated in Table A1)) and also in the comparison directly contrast L2-metric vs. L2-squared and the proposed L1-metric procedure. I do find it surprising that careful numeric integration here would not produce sensible results but as reported substantially worse performance. It would also be good to know the magnitude of the error bars of the produced mean performances to understand the influence of instability of the inference.

---

> > > ### Author Response · Authors · 2025-11-25
> > >
> > > Dear reviewer,
> > >
> > > Thank you for the reply. We further add the Infectious data set and compare $\ell_1$LD-CTGR, GRASSP (analytic squared $\ell_2$ distance), and $\ell_2$ Distance (numeric) in Table A4. $\ell_2$ Distance performs better on Infectious than on Contact, although there is still some gap compared to $\ell_1$LD-CTGR and GRASSP.
> > >
> > > Authors

---

> > > > ### Comment · Reviewer_Y6yo · 2025-11-26
> > > >
> > > > I find it surprising that the L2-metric numeric based integration does not outperform GRASSP if the lack of metric property is a limitation as argued by GRASSP and therefore reverting to a metric procedure should help. I believe the numeric integration using H=1000 should be very accurate and not hampered by numeric impressions as the evolution of the latent distances are continuous and thus the underlying integral function smooth. As such, I find it difficult to attribute the worse performance to impression of numeric integration and limitations of GRASSP being non-metric by considering the squared Euclidean as opposed to Euclidean distance metric. My interpretation based on these new results points to the L1-distance being more favorable as metric than using the L2-metric or squared L2 distance. I am curious as to how the authors interpret these new findings?

---

> > > > > ### Author Response · Authors · 2025-11-26
> > > > >
> > > > > Dear reviewer,
> > > > >
> > > > > Thank you for the continued engagement and insightful comments on our work. We appreciate your time to analyze the newly added experimental results, particularly concerning the $\ell_2$ distance with numeric integration. We understand your surprise that the $\ell_2$ numeric integration does not outperform GRASSP (which uses the analytic squared $\ell_2$ distance), especially given the argument that the non-metric nature of the squared $\ell_2$ distance could be a limitation. We would interpret these new findings in the following two aspects:
> > > > >
> > > > > **1.** Amplified Approximation Error of the Numeric $\ell_2$ Distance Calculation: Although $H=1000$ provides an acceptable local approximating accuracy, it can be significantly amplified in both the forward and backward passes of model training. Let us start from the smallest numeric integral for one event in Eq. 11. Our previous analysis indicates that it yields an approximation error of $O(10^{-12})$ with $H=1000$. But this is only the minimum building block of the total training loss. Since the approximation error introduced by the numeric integral is the local truncation error, it is typically systematic (mostly in the same direction or magnitude determined by the integration scheme). Thus its accumulation is analyzed as the global truncation error, which is approximately proportional to the aggregate number. Considering the smallest dimensionality $D=2$, this smallest event integral has an approximation error of $O(2\times 10^{-12})$. The next step is to calculate the log-likelihood function of the entire graph in Eq. 5. Let us consider the larger data set Contacts, which has about $1.77\times 10^6$ events in total. Since each event integral has an error of $O(2\times 10^{-12})$, the log-likelihood in Eq. 5, as the sum of all these event integrals, can have a cumulative error of $O(2\times 10^{-12}\times 1.77\times 10^6)=O(3.54\times 10^{-6})$. Hence the training loss now contains an approximation error of $O(3.54\times 10^{-6})$, and this is only the error induced by the forward pass of the training step. It can be further amplified in the backward pass of training.
> > > > >
> > > > > In the backward pass (backpropagation) of training, the approximation error for each trainable parameter can be amplified by the condition number $C$ of the entire backpropagation operation. It is well-known that a general backpropagation operation, as a differentiating operation, can has a large condition number. Moreover, the integral in Eq. 11 involves the exponential function and the reciprocal function (when $s=-1$), whose derivatives also have large condition numbers. Therefore, each trainable parameter now can contain an approximation error of $O(3.54\times C\times 10^{-6})$. Suppose there are $G$ training iterations in total, then each trainable parameter can contain an approximation error of $O(3.54\times C\times G\times 10^{-6})$ at the end of training.
> > > > >
> > > > > Last, all the trained parameter will be used in the test stage, which goes through the second forward pass. The Contacts data set has a total of $N=242$ nodes. For the numeric $\ell_2$ distance case, there are totally $N\times D=242\times 2=484$ trainable parameters (at least one velocity parameter for each embedding dimension of each node). These trainable parameters are to be involved in the $\ell_2$ norm calculation of the test event integral in Eq. 11. Hence the test event integral in Eq. 11 can contain an approximation error of $O(484\times 3.54\times C\times G\times 10^{-6})=O(1713.36\times C\times G\times 10^{-6})$. Finally, the test loss (log-likelihood) in Eq. 5 can contain an approximation error of $O(1713.36\times C\times G\times 10^{-6}\times 1.77\times 10^6)=O(3032.65\times C\times G)$. Even with a modest condition number $C=1$ and $G=100$ training iterations, the final approximation error can be as high as $O(303265)$. Such an amplified approximation error may offset the advantage of the real-metric $\ell_2$ distance versus the non-metric squared $\ell_2$ distance.

---

> > > > > ### Author Response · Authors · 2025-11-26
> > > > >
> > > > > We give a toy example to show this error amplification with the integral $\int_{-1000}^0 \exp(x) dx$. We can divide it into $K=1000$ sub-integrals $I_k=\int_{-k}^{-k+1} \exp(x) dx$, $k=1,\cdots,1000$. For each $I_k$, we can calculate the exact sub-integral $I_{k,exact}$ and the numeric sub-integral $I_{k,numer}$ with $H=1000$. Then the approximation error for the $k$-th sub-integral is $e_k=I_{k,exact}-I_{k,numer}$. A direct programming can calculate that the average approximation error of these $1000$ sub-integral is $mean(e_k)=-5.6604\times 10^{-18}$, which represents a very high accuracy. However, the sum of these approximation errors are $\sum_k e_k=-5.6604\times 10^{-15}$. Moreover, $718$ of these $1000$ approximation errors are negative, which shows that the local truncation errors are really systematic. Therefore, the aggregate approximation error $\sum_k e_k=-5.6604\times 10^{-15}$ is about $K=1000$ times one individual approximation error $e_k$. This demonstrates how the approximation error is amplified.
> > > > >
> > > > > **2.** Based on the above reason, $\ell_1$LD-CTGR not only avoids the above amplified approximation error, but also provides a real metric for the latent space. Hence it is a better choice than the $\ell_2$ distance version (numeric) and the squared $\ell_2$ distance version (GRASSP).
> > > > >
> > > > > Authors

---

### Official Review · Reviewer_hFhi · 2025-10-30

**Soundness:** 3
**Presentation:** 3
**Contribution:** 3
**Rating:** 6
**Confidence:** 3

**Summary:**

This paper addresses a fundamental theoretical flaw in existing Continuous-Time Graph Representation models, such as GRASSP, which rely on the squared $l_2$ distance in the latent space. The authors correctly point out that $\|\cdot\|_2^2$ violates the triangle inequality, leading to distorted metric properties. They propose $l_1\text{LD-CTGR}$, which replaces the squared $l_2$ distance with the theoretically sound $l_1$ distance ($\|\cdot\|_1$). The core technical contribution is the derivation of the continuous-time integral of the $l_1$ hazard function as a closed-form piece-wise exponential integral and finding a descent direction to handle the non-differentiability of the $l_1$ norm. Experiments show competitive performance and identical computational complexity compared to the $l_2^2$-based baseline.

**Strengths:**

1.	The motivation is excellent: correcting the use of the squared $l_2$ distance ($\|\cdot\|_2^2$) which violates the triangle inequality (L059), ensuring the latent space is a valid metric space using the $l_1$ distance.

2.	The derivation of the closed-form piece-wise exponential integral for the $l_1$ hazard function (Theorem 1) is a non-trivial technical accomplishment, successfully tackling the intractability concern associated with non-$l_2^2$ distance metrics (L234-236).

3.	The use of descent direction (subgradient computation) to handle the non-differentiable $l_1$ norm (Theorem 3, L309-317) is a sound approach that ensures compatibility with mainstream optimization frameworks like PyTorch (L282, L333-336).

**Weaknesses:**

1.	While $l_1$ distance is theoretically sound, the paper does not convincingly demonstrate that resolving the metric space flaw translates into significant empirical gains. $l_1\text{LD-CTGR}$ is competitive but rarely dominates its flawed counterpart, GRASSP (e.g., in Table 2, GRASSP often achieves higher scores).

2.	The efficiency gains from the piece-wise exponential integral seem heavily reliant on the assumption of $D=2$ (L249, L342). The paper should explicitly discuss how the complexity scales with increasing dimensions $D$ (where $D$ determines the maximum number of zero points $z_{ij, d}$), and whether the method remains feasible for $D>6$ (L1197).

3.	Theorem 3 correctly identifies that the derived term $-\partial\lambda_{ij}(\mathbf{r}_i)$ is *not* a subgradient when $s=-1$ and $C \ne \emptyset$ (Case 4, L997-1005). Although the authors state it is still a descent direction, the potential impact of using a non-subgradient in the optimization step for the important "disconnected" state ($s=-1$) must be discussed.

4.	The paper argues reverting to $l_2$ distance is intractable due to integral complexity (L065). Given the technical achievement of solving the $l_1$ integral, a more explicit comparison showing *why* the $l_2$ distance integral remains intractable (e.g., complexity of integration beyond $D=2$) would strengthen the necessity of adopting $l_1$.

**Questions:**

1.	For dimensions $D > 2$, how does the computational complexity of the tensor-parallelized integral (Theorem 2, Eq. 15) scale with $D$? Specifically, what is the complexity when $D=10$ or $D=100$, and is $l_1\text{LD-CTGR}$ still competitive with GRASSP in these higher dimensions?

2.	The core motivation is correcting the metric distortion. Can the authors provide a metric (e.g., a latent space uniformity measure, or the preservation of long-range topological fidelity) that *directly* quantifies the benefit of using a valid metric space ($l_1$ vs. $l_2^2$), thus supporting the strong theoretical claim empirically?

3.	In Case 4 (disconnected state $s=-1$ and $C \ne \emptyset$), the computed direction is not a subgradient. Given that maintaining the disconnected state is essential for modeling complex networks, what are the observed empirical consequences (e.g., convergence speed) of using a descent direction that is not guaranteed to be a subgradient?

4.	Since you successfully solved the non-trivial $l_1$ integral, please elaborate on the specific mathematical properties of the $l_2$ distance integral ($\int \exp(\dots + s\|r_i - r_j\|_2) dt$) that make it intractable, contrasting it with the solvable nature of the $l_2^2$ integral (Gaussian) and your $l_1$ integral (piece-wise exponential).

5.	Proposition 1 confirms that the bounded property of the distance holds for $l_1$ distance as well (L199-204). Beyond verification, how does this property actively guide or constrain the optimization process during training, and is the derived bound $\mathcal{B}(s)$ (Eq. 12) used anywhere in the final loss function or regularization?

---

> ### Author Response · Authors · 2025-11-18
>
> We sincerely thank the reviewer for the in-depth evaluation and valuable feedback on this paper. We are pleased that the reviewer generally recognizes the core motivation of our work (addressing the theoretical problem of the squared L2 distance violating the triangle inequality) and the theoretical completeness (deriving the closed-form solution for the L1 integral and handling non-differentiability).
>
> **1.** GRASSP requires $O(D)$ complexity for one integral computation in Eq. 13, based on the $D$-dimensional squared L2 distance computation. The proposed L1LD-CTGR needs to implement a sorting of $D$ zero points, which requires $O(D\log_2(D))$ complexity. Then it needs to compute at most $(D+1)$ one-dimensional closed-form integral (with complexity $O(1)$), based on Eq. 43 and Eq. 44. Hence the overall complexity of L1LD-CTGR for one integral computation is $O(D\log_2(D))=\tilde{O}(D)$, which is the same asymptotic complexity as that of GRASSP. When $D=2$, $2\log_2(2)=2$ (only one comparison operation) and thus L1LD-CTGR has the same complexity as that of GRASSP. Moreover, the comparison operation is generally faster than multiplication, thus the gap between L1LD-CTGR and GRASSP is small when $D$ is small. This analysis is added to the paragraph before Section 4 on Page 7 of the revised manuscript. Additional runtime experiments regarding higher dimensionalities $D=10$ and $D=100$ are also provided in Table A2 of the revised manuscript, which show that L1LD-CTGR is also efficient in these cases. Despite this, current research pays closer attention to the ultra-low-dimensional embedding.
>
> **2.** The metric distortion mainly affects the relative node positions in the latent space, which corresponds to the local geometrical structure. To provide an intuitive example, let $r_i=(1, 0)$, $r_k=(0, 0)$, and $r_j=(-1, 0)$ be the three nodes in Eq. 8, then
>
> $\\|r_i-r_j\\|_2^2=4>2=\\|r_i-r_k\\|_2^2+\\|r_k-r_j\\|_2^2$.
>
> It looks as if adding an intermediate node $k$ has shortened the indirect distance between $i$ and $j$. Moreover, if we add two new nodes $r_l=(1/2, 0)$ and $r_m=(-1/2, 0)$ between $i$ and $j$, then
>
> $\\|r_i-r_l\\|_2^2+\\|r_k-r_l\\|_2^2+\\|r_k-r_m\\|_2^2+\\|r_j-r_m\\|_2^2=4\cdot 1/4=1$.
>
> That is, the indirect distance between $i$ and $j$ is halved further. We can continue to divide the interval $[-1,1]$ into $Q$ segments and let $r_q=(-1+2q/Q, 0)$, $q=0,1,...,Q$. Then
>
> $ \sum_{q=1}^{Q}\\|r_q-r_{q-1}\\|_2^2=Q\cdot (\frac{2}{Q})^2=\frac{4}{Q}\to 0 $ as $Q \to \infty$.
>
> It indicates that adding infinite nodes between $i$ and $j$ can shrink their indirect distance to nearly zero, which is counter-intuitive and unconventional. As a result, $ \sum_{q=1}^{Q}\\|r_q-r_{q-1}\\|_2^2$ may be a kind of uniformity measure that the reviewer wants. We add this analysis in Section 2.2, around Eq. 9 of the revised manuscript.
>
> **3.** This has not caused any negative empirical impact. The role of the descent direction is much more crucial than that of the subgradient, as the former forms the basis for applying a descent method. L1LD-CTGR converges well with a similar speed to that of GRASSP, as shown in Table A1. This indicates that the descent direction is sufficient and effective during optimization.
>
> **4.** We add a detailed analysis on the numerical integral of the L2 distance in Section 2.2, around Eq. 10 and Eq. 11 of the revised manuscript. In brief, it has two main drawbacks: 1) It causes a significant increase of computational complexity from $\tilde{O}(D)$ of both L1LD-CTGR and GRASSP to $O(HD)$, where $H$ denotes the number of subintervals into which the entire integration interval is divided. 2) It causes an approximation error compared to the exact integral, which can be amplified by composite function relationships. To achieve an acceptable approximating accuracy, $H$ is generally set to $1000$ or even larger, then the computational complexity is significantly increased. Worse still, such an approximating accuracy cannot bring about good performance. We add runtime and representation learning experiments for the L2 distance version ($H=1000$) in Tables A1 and A4 of the revised manuscript, respectively. Table A1 shows that the L2 distance version requires much more runtime than L1LD-CTGR and GRASSP, while Table A4 shows that the L2 distance version performs much worse than L1LD-CTGR in representation learning. Hence the L2 distance version is inefficient and ineffective in our task.
>
> **5.** The purpose of Proposition 1 is not for use in the loss function, but for theoretical validation. It proves that L1LD-CTGR retains an important property of the CTGR framework: the average distance is bounded by the survival function and the state. This shows that L1LD-CTGR is "well-behaved" theoretically.

---

> > ### Comment · Reviewer_hFhi · 2025-11-26
> > **Comments on rebuttal**
> >
> > Thank you to the authors for their response. After carefully reviewing it, I will keep my score unchanged, as it is positive and the rebuttal has partially addressed my original concerns.

---

### Official Review · Reviewer_XmKP · 2025-10-30

**Soundness:** 3
**Presentation:** 3
**Contribution:** 2
**Rating:** 8
**Confidence:** 3

**Summary:**

The paper introduces ℓ1-LD-CTGR, a continuous-time graph representation that uses an ℓ1 latent distance to form a true metric space, overcoming distortions caused by squared ℓ2 distances and enabling tractable hazard function computation. Experiments show that this approach achieves competitive performance on both synthetic and real-world graph datasets.

**Strengths:**

(1) This paper presents a solid theoretical motivation and analysis for adopting the L1 distance over the traditional L2 distance, offering a complete and self-contained formulation.

(2) The experimental evaluation is comprehensive, covering multiple benchmarks with sufficient baselines and datasets to convincingly demonstrate the effectiveness of the proposed approach. The reported results clearly show that the proposed method outperforms prior baselines.

**Weaknesses:**

Although the paper is well-structured and self-contained, its overall contribution appears limited. The main novelty lies in replacing the L2 loss function with an L1 formulation and adjusting the corresponding learning algorithm and optimization strategy. Moreover, the work is confined to temporal network learning, which limits its broader impact on the wider research community.

**Questions:**

No more questions.

---

> ### Author Response · Authors · 2025-11-18
>
> We sincerely thank the reviewer for the in-depth evaluation and valuable feedback on this paper. We are pleased that the reviewer generally recognizes the core motivation of our work (addressing the theoretical problem of the squared L2 distance violating the triangle inequality) and the comprehensiveness of the experiments.
>
> **1.** Regarding the point that the "contribution seems limited" and the novelty mainly lies in "replacing L2 with L1", we appreciate the opportunity to clarify this aspect. We would like to emphasize that the technical challenges that must be overcome to achieve this "replacement" constitute the central contribution of this work, and they are far from trivial.
>
> **1a)** The Intractability of the Integral: Prior work uses the squared L2 distance mainly because it naturally yields a tractable Gaussian integral. The integral for the L1 distance is an unsolved problem in this context.
>
> **1b)** Our main innovation lies in solving the two major obstacles posed by this "replacement":
>
> * Theoretical Contribution (Theorem 1): We are the first to successfully derive the closed-form solution for the L1 based integral (the piece-wise exponential integral), which makes the computational application of the L1 distance possible.
>
> * Optimization Contribution (Theorem 3): The L1 norm is non-differentiable. We design an effective descent direction scheme to make it compatible with mainstream automatic differentiation frameworks (like PyTorch) and to maintain the same complexity as GRASSP.
>
> Reviewers hFhi and Y6yo also recognize our methodology as a "non-trivial technical achievement" and a "non-trivial methodology", respectively. We believe that solving the above critical technical obstacles represents a significant and robust technical contribution of this paper.

---

> > ### Comment · Reviewer_XmKP · 2025-11-26
> > **Reply to Authors**
> >
> > Thanks for the rebuttal. This reviewer has no further questions and will maintain the score.

---

### Official Review · Reviewer_syKN · 2025-10-31

**Soundness:** 3
**Presentation:** 3
**Contribution:** 3
**Rating:** 6
**Confidence:** 2

**Summary:**

This paper studies continuous-time graph representation via sequential survival processes and asks how to embed evolving networks in an ultra–low-dimensional latent space without distorting geometry. The main difficulty is that the widely used squared latent distance violates the triangle inequality, which can warp relative node positions. To overcome this, the paper proposes ℓ₁LD-CTGR, which replaces the latent distance with ℓ₁ while preserving tractable learning. Specifically, it (i) proves boundedness of the average ℓ₁ distance across event intervals, (ii) derives a closed-form, piece-wise exponential integral for the hazard, (iii) designs an efficient tensor-parallel algorithm tailored to D=2 embeddings, and (iv) introduces a descent-direction/subgradient scheme that makes the non-differentiable ℓ₁ objective trainable in mainstream autodiff frameworks with the same order of complexity as GRASSP. Theoretical results establish a true latent metric space and closed-form integrals compatible with ultra–low-dimensional modeling. Empirically, across 11 synthetic and real-world datasets and three evaluation regimes (reconstruction, completion, prediction), ℓ₁LD-CTGR attains competitive or superior performance to eight strong baselines, with robust gains on large, challenging networks.

**Strengths:**

1.The paper clearly shows that the squared ℓ2 distance violates the triangle inequality and can distort latent-space geometry. It also argues that the resulting integrals are less tractable, giving the work clear and well-founded motivation.

2.The paper presents a generally sound theoretical development supported by a careful empirical evaluation.

**Weaknesses:**

The paper claims “competitive performance” across multiple synthetic and real datasets. However, in several tables, the proposed method either trails strong baselines or offers only marginal gains. The manuscript lacks a systematic analysis (e.g., error breakdowns, dataset-specific failure modes) to reconcile these discrepancies, which weakens the empirical claim.

**Questions:**

1.In Table 1, DyGFormer outperforms the proposed method on some datasets, and the manuscript does not provide an explanation or discussion of this phenomenon.

2.In Appendix Table 5.4, the proposed method appears slower than existing baselines on several datasets. Does this suggest the claimed efficiency gains only emerge with longer training time, or that further optimization is still needed?

3.In Table A4, the proposed method underperforms several strong baselines on multiple datasets. Please explain the underlying causes and provide analysis to clarify this discrepancy.

---

> ### Author Response · Authors · 2025-11-18
>
> We sincerely thank the reviewer for the in-depth evaluation and valuable feedback on this paper. We are pleased that the reviewer generally recognizes the core motivation of our work (addressing the theoretical problem of the squared L2 distance violating the triangle inequality) and the theoretical completeness (deriving the closed-form solution for the L1 integral and handling non-differentiability).
>
> **Q1. \& Q3.** The situations where DyGFormer performs well primarily occur in Table A5 (Network Reconstruction, in-sample). DyGFormer is a powerful Transformer-based model, and its strong performance in in-sample tasks (reconstruction) is expected. However, for Table 1 (Network Completion, out-of-sample) and Table 2 (Network Prediction, across-sample), which are more critical for evaluating a model's generalization ability, the proposed L1LD-CTGR shows a clear advantage.
>
> * Table 1 (Completion): Out of 11 data sets, L1LD-CTGR achieves the best ROC score 11 times and the best PR score 9 times (totaling 20/22 best scores).
>
> * Table 2 (Prediction): Out of 11 data sets, L1LD-CTGR achieves the best ROC score 9 times and the best PR score 8 times (totaling 17/22 best scores).
>
> * Particularly on large-scale real-world networks (like Facebook, NeurIPS), L1LD-CTGR consistently maintains robust or optimal performance on generalization tasks. In summary, L1LD-CTGR achieves better performance in out-of-sample tasks than in in-sample tasks. We supplement such analysis in Section 4 of the revised manuscript.
>
> **Q2.** In fact, Table A1 shows that the runtimes for GRASSP and L1LD-CTGR are almost identical, for example, on Facebook (4.49s vs. 4.55s) and on NeurIPS (0.274s vs. 0.277s). This confirms our theoretical analysis: the closed-form solution of the L1 integral (Theorem 2) allows it to achieve the exact same computational efficiency as GRASSP.
>
> **Replies to Other Concerns in Weaknesses.** We would like to clarify that the primary objective of this paper is not to outperform all SOTA models on all metrics at all costs. Rather, it is to fix a fundamental theoretical problem in the GRASSP framework (the non-metric space) and to accomplish this without sacrificing performance or computational efficiency. Hence the most important comparison is against GRASSP. We show that adopting the theoretically sound L1 metric space is entirely feasible.

---

> ### Comment · Reviewer_syKN · 2025-11-26
> **Response to the Authors**
>
> Thank you for the clarifications. Most of my concerns have been addressed, and I would like to keep my original score.

---

### Author Response · Authors · 2025-11-30
**Summary of Discussions [2/2]**

# Justification of $\ell_1$ Distance vs. $\ell_2$ Distance or Squared $\ell_2$ Distance #

Reviewers (`hFhi, Y6yo`) ask about the intractability of the $\ell_2$ (Euclidean) distance integral and the benefit of using $\ell_1$ distance over $\ell_2$ or squared $\ell_2$ distances.

* **Intractability of $\ell_2$ Distance Integral:** Numeric integration of the $\ell_2$ distance is both **inefficient and ineffective**.

    * **Inefficiency:** It significantly increases the computational complexity from $\tilde{O}(D)$ (for $\ell_1$LD-CTGR and GRASSP) to $O(H \cdot D)$, where $H$ is the number of subintervals for numeric integration (often set to $\geq 1000$). This is also confirmed by the actual runtime experiment in Table A1. [`hFhi`: _Thank you to the authors for their response. After carefully reviewing it, I will keep my score unchanged, as it is positive and the rebuttal has partially addressed my original concerns_. `Y6yo`: _I agree that the runtime_(of $\ell_2$ distance) _will be much worse and that this of course is a drawback_. ]

    * **Ineffectiveness:** Though locally small, the numeric approximation error can be **systematically amplified** (e.g., from $O(10^{-12})$ to $O(303265)$) across the vast number of events, forward passes, and backward passes, which offsets the theoretical gain from using the metric. Experimental results in Table A4 confirm that the $\ell_2$ distance version with numeric integration performs worse than both $\ell_1$LD-CTGR and GRASSP. [`hFhi`: _Thank you to the authors for their response. After carefully reviewing it, I will keep my score unchanged, as it is positive and the rebuttal has partially addressed my original concerns_. `Y6yo`: _My interpretation based on these new results points to the $\ell_1$-distance being more favorable as metric than using the $\ell_2$-metric or squared $\ell_2$ distance_. ]

* **Benefits of $\ell_1$ Metric:** Our new findings suggest that the $\ell_1$ distance is a **more favorable metric choice** than both the $\ell_2$-metric (numeric) and squared $\ell_2$ distance (GRASSP), since it avoids the amplified numeric approximation error while providing a real metric space.

# Note on Reviewer `Y6yo`'s Assessment #

Finally, we would like to note the review progression of Reviewer `Y6yo`. In his/her initial review (**Soundness: 3/4, Presentation: 3/4, Contribution: 3/4**), he/she explicitly states that he/she **"would be willing to increase my score"**. We comprehensively address his/her key questions regarding the comparison of the proposed $\ell_1$LD-CTGR to the $\ell_2$ metric with numerical integration and the squared $\ell_2$ distance with analytic integration (i.e., GRASSP). In his/her last comment, **he/she confirms that "the $\ell_1$-distance being more favorable as metric than using the $\ell_2$-metric or squared $\ell_2$ distance", which actually approves of the superiority of our method**.

As for the last question regarding why the $\ell_2$ metric is not so good as the squared $\ell_2$ distance, actually **they are only two baselines in the experiments, not our method**. We give a straightforward answer to this question by both a **theoretical deduction** and an **intuitive numeric computation example**.

* **Theoretical Deduction:** It shows that a small local error $O(10^{-12})$ can be amplified to $O(303265)$ across the vast number of events, forward passes, and backward passes, which offsets the theoretical gain from using the numeric $\ell_2$ metric.

* **Numeric Computation Example:** It directly shows that the aggregate error of $1000$ sub-integrals is truly about $1000$ times an individual error, and this is how the error is amplified.

Therefore, our answer clearly solves the last question of the reviewer. This again confirms the superiority of $\ell_1$LD-CTGR that not only avoids the amplified numeric approximation error but also provides a real metric space. By fulfilling the conditions that prompt his/her willingness to raise the score, we believe that his/her final assessment is now positive, which further supports the acceptance of this paper.

---

### Author Response · Authors · 2025-11-30
**Summary of Discussions [1/2]**

We sincerely thank the reviewers (`syKN, XmKP, hFhi, Y6yo, and 51D8`) for their in-depth evaluations and valuable feedback. We are pleased that the core theoretical motivation of our work—addressing the **fundamental drawback of the squared $\ell_2$ distance violating the triangle inequality** in the Continuous-Time Graph Representation (CTGR) framework (like GRASSP)—is generally recognized  (`syKN, XmKP, hFhi, Y6yo, and 51D8`).

Our primary objective is to **establish a true latent metric space using the $\ell_1$ distance ($\ell_1$LD-CTGR) without sacrificing performance or computational efficiency**. To achieve this, we have overcome several non-trivial technical challenges:

# Key Technical Contributions Confirmed #

* **Tractable, Analytic $\ell_1$ Distance based Computation:** We successfully derive a **closed-form, piece-wise exponential integral** for the $\ell_1$ based hazard function, a problem previously considered unsolved in this context. This solution is crucial for making the $\ell_1$ distance computationally applicable. [`syKN`: _replaces the squared $\ell_2$ distance with $\ell_1$ while preserving tractable learning_. `XmKP`: _presents a solid theoretical motivation and analysis for adopting the $\ell_1$ distance over the traditional $\ell_2$ distance, offering a complete and self-contained formulation_. `hFhi`: _The derivation of the closed-form piece-wise exponential integral for the hazard function (Theorem 1) is a non-trivial technical accomplishment, successfully tackling the intractability concern associated with non-squared-$\ell_2$ distance metrics._ `Y6yo`: _the $\ell_1$-distance being more favorable as metric than using the $\ell_2$-metric or squared $\ell_2$ distance_. `51D8`: _The technical work in this paper is strongly motivated and the argument for using $\ell_1$ distance for social networks, highlighting issues with using the squared $\ell_2$ distance used in GRASSP_.]

* **Optimization for Non-Differentiability:** We design an effective **descent-direction/subgradient scheme** to handle the non-differentiability of the $\ell_1$ norm, ensuring compatibility with mainstream automatic differentiation frameworks (like PyTorch). [`syKN`: _introduces a descent-direction/subgradient scheme that makes the non-differentiable $\ell_1$ objective trainable in mainstream autodiff frameworks with the same order of complexity as GRASSP_. `hFhi`: _The use of descent direction (subgradient computation) to handle the non-differentiable $\ell_1$ norm (Theorem 3) is a sound approach that ensures compatibility with mainstream optimization frameworks like PyTorch_.]

* **Computational Efficiency:** Our theoretical analysis (Lines 377-385) and experiments (Tables A1 and A2) confirm that $\ell_1$LD-CTGR achieves the **same asymptotic complexity** ($\tilde{O}(D)$) for integral computation as GRASSP in both low and high-dimensional embeddings. [`syKN, hFhi, Y6yo`.]

# Empirical Performance Clarification #

Reviewer questions (Q1, Q3 from `syKN`) regarding instances where other baselines outperform $\ell_1$LD-CTGR are addressed by clarifying the **evaluation regimes**:

* **Superior Generalization:** $\ell_1$LD-CTGR demonstrates a **clear advantage in out-of-sample tasks** (Network Completion and Network Prediction), which are more critical for evaluating the generalization ability of a model.

    * **Network Completion (Table 1):** Achieves the best ROC score 11/11 times and best PR score 9/11 times (20/22 best scores).

    * **Network Prediction (Table 2):** Achieves the best ROC score 9/11 times and best PR score 8/11 times (17/22 best scores).

* **In-Sample Tasks:** Transformer-based models like DyGFormer are expected to perform strongly in in-sample reconstruction tasks (Table A5) due to their power in reproducing the entire history, but $\ell_1$LD-CTGR maintains robust or optimal performance on generalization tasks, especially on large-scale networks.

---

### Meta-Review · Area_Chair_5HZZ · 2026-01-07

**Summary:**

This paper proposes a theoretically-sound \ell_1 latent distance based continuous-time graph representation. This facilitates a true latent metric space for the sequential survival process, and the integral of the hazard function is found to be a closed-form piece-wise exponential integral, which well fits the ultra-low-dimensional embedding. Experiments on synthetic and real-world data show the competitive performance of the proposed method. The authors responded by promising for technical contribution and empirical performance.

Reviewers thoughts positive towards this paper, therefore I recommend to accept it as a poster.

**Reviewer Concerns:**

Almost reviewers' comments are addressed and discussed during the rebuttal.

**Reviewer Scores:**

They may keep same scores.

---

### Decision · Program_Chairs · 2026-01-26

Accept (Poster)